# *Rubus coreanus* extract prevents kidney fibrosis through TGF-β/Smad pathway inhibition

**Wencheng Jin[1,2]◉, Ara Cho[3]◉, Bogyeong Cho[2], Nayeon Shin[2], Yun Kyu Oh[1,2], Chun Soo Lim[1,2], Jung Pyo Lee[1,2], Jeonghwan Lee [ID][1,2]***

**1** Department of Internal Medicine, College of Medicine, Seoul National University, Seoul, Korea,
**2** Department of Internal Medicine, Seoul National University Boramae Medical Center, Seoul, Korea,
**3** Translational Medicine Major, Seoul National University College of Medicine, Seoul, Korea

◉ These authors contributed equally to this work.
* jeonghwan@snu.ac.kr

## Abstract

*Rubus coreanus* has been found to have various health benefits including anti-oxidative effects. In this study, we aimed to investigate the efficacy of water-soluble extract of *Rubus coreanus* (MN705) in preventing kidney fibrosis in a mouse model of unilateral ureteral obstruction (UUO) and in an *in vitro* model of TGF-β-challenged HK-2 cells. Male C57BL/6 mice (7 weeks old) were randomly assigned to the sham/vehicle (distilled water), sham/MN705 (600 mg/kg/day), UUO/vehicle, UUO/MN705-low dose (300 mg/kg/day), and UUO/MN705-high dose (600 mg/kg/day) groups. After 7 days of pre-treatment, sham or UUO operation was performed, and treatment drugs were administered at the same dose for 7 days. In addition, HK-2 cells and human renal proximal tubular epithelial cells were cultured and challenged with TGF-β (2 ng/ml) with or without an extract of *Rubus coreanus* (0.05–0.2 mg/ml). In the histopathological specimen of Masson's trichrome stain, areas of kidney interstitial fibrosis were attenuated in the treatment group (10.6±1.3% area vs. 17.2±2.3% area, $P < 0.001$). In the western blot analysis, protein abundance of α-SMA (12.9±5.9 vs. 19.8±3.1 fold changes compared to sham group, $P = 0.046$) significantly decreased in the treatment group. In the *in vitro* experiment, HK-2 cells treated with TGF-β and MN705 showed a dose-dependent significant decrease in the protein expression of fibronectin and phospho-Smad2/3 with increase of MnSOD. The extract of *Rubus coreanus* attenuates kidney fibrosis in the UUO mouse model and TGF-β-treated human kidney proximal tubular cells. TGF-β-related Smad and Smurf signaling pathways involved in the development of fibrosis are effectively inhibited through extract of *Rubus coreanus* and can be a potential target for prevention of kidney fibrosis.

## Introduction

Kidney fibrosis is a pathological process involving the accumulation of extracellular matrix in the interstitium that is a common process of nearly all progressive chronic kidney diseases [1]. Mild kidney injury may initially lead to the deposition of the

**Data availability statement:** All relevant data are within the manuscript and its Supporting Information files.

**Funding:** This study was supported by a multidisciplinary research grant-in-aid from the Seoul Metropolitan Government, Seoul National University (SMG-SNU) Boramae Medical Center (No. 04-2023-0025). The funders had no role in study design, data collection and analysis, decision to publish, or preparation of the manuscript. Boram Weon, Yun Kyu Oh, Chun Soo Lim, Jung Pyo Lee, and Jeonghwan Lee are employed by SMG-SNU Boramae Medical Center, which provided research funding, and receives regular salary payments.

**Competing interests:** Medvill Korea provided MN705 and other natural substances used in this study free of charge for academic research purposes. However, the company did not participate in the study design, data analysis, or manuscript preparation. Additionally, we declare that there are currently no patents, products in development, or marketed products associated with this research. We confirm that these competing interests do not alter our adherence to PLOS One policies on sharing data and materials.

fibrous matrix, which can facilitate tissue repair and be subsequently absorbed [2]. However, in chronic kidney disease, excessive deposition of the fibrous matrix can lead to the destruction of kidney parenchyma, loss of nephron, progressive deterioration of kidney function, and ultimately, kidney failure [3]. Unfortunately, the cure for kidney fibrosis is currently unknown, and available treatment strategies rely on the blockade of the renin-angiotensin system and blood glucose control, which can only slow the progression of chronic kidney disease and kidney fibrosis delaying the development of end-stage kidney disease [4].

The rediscovery of bioactive compounds with broad effects and fewer side effects has become a growing area of interest, and natural herbal products have been found to have great potential for treating kidney fibrosis [5]. Therefore, developing new drugs based on natural substances to alleviate kidney fibrosis more effectively is necessary [6]. *Rubus coreanus* is a wild berry native to eastern Asia that has been used as a traditional beverage and healthy food for millennia. Previous studies have shown that *Rubus coreanus* can enhance immunity [7]. Moreover, it can promote lipolysis and thermogenesis via increased adipose tissue lipolysis [8]. Other studies have shown that *Rubus coreanus* can save the bone loss of diabetes-induced osteoporosis [9].

Based on this background data, we explored candidate substances with the protective effect of kidney fibrosis among extracts of natural herbal products. After the screening, *Rubus coreanus* (MN705) was selected as a potential substance attenuating kidney fibrosis. In this study, we aimed to investigate the efficacy of *Rubus coreanus* in inhibiting kidney fibrosis in both *in vivo* mice unilateral ureteral obstruction (UUO) model and *in vitro* transforming growth factor-β (TGF-β) challenged HK-2 cells model. In addition, we aim to determine the underlying mechanism of anti-fibrosis through *Rubus coreanus*.

## Materials and methods

### Selection of natural product extract for anti-fibrosis

Nine traditional Chinese herbal medicines and health-promoting natural substances were selected and extract powders were provided by pharmaceutical company (Medvill Co., Ltd. Korea). The nine candidate natural substances were selected based on plant-derived materials traditionally recognized for their beneficial effects on physical health. These natural substances were processed into powder form through hot water extraction, concentration, and freeze-drying, and subsequently utilized in the experiment. To conceal the names of the materials, each of the materials was assigned a serial number from MN701 to MN709. The cellular activity of individual substances was checked using the MTT assay method, and substances that maintained cell viability relatively well in the general therapeutic dose range were selected as candidates for anti-fibrosis effects. 3-(4,5-Di-2-yl)-2,5-ditetrazolium bromide (MTT) assay was performed to evaluate the cytotoxic effect of ruscogenin. The cells were seeded in 96-well plates ($2 \times 10^3$ cells/well), cultured for 48 h, and treated with ruscogenin (0.001–100 μM). At the end of the incubation period, the cells were incubated with

1 mg/ml MTT solution. After three hours, the absorbance was measured at 450 nm and the data were assessed using an ELX-800 spectrometer reader (Bio-Tek Instruments).

## Experimental animals and establishment of the *in vivo* model

Wild-type (WT) mice (C57BL/6) were obtained from Koatech (Seoul, Korea). The mice that were used in the experiments were raised in a pathogen-free facility and they were 7 weeks old when subjected to experimentation.

Male C57BL/6 mice (7 weeks old) were randomly assigned to the sham/vehicle (distilled water, n = 4), sham/MN705 (600 mg/kg/day, n = 4), UUO/vehicle (n = 6), UUO/MN705-low dose (300 mg/kg/day, n = 6), and UUO/MN705-high dose (600 mg/kg/day, n = 6) groups. Pretreatment with MN705 or vehicle started 1 week before sham/UUO operation. Different concentrations of MN705 (300 mg/kg/day or 600 mg/kg/day) were administered orally using a stainless feeding needle to the mice. Equivalent volume of distilled water was administered as vehicle in the control group. The *in vivo* kidney fibrosis model was established by inducing unilateral ureteral obstruction (UUO) in male C57BL/6 mice. Briefly, the mice were anesthetized with an intraperitoneal injection of Rompun™ (xylazine 10 mg/kg; Bayer Korea Co., Ansan, Korea) mixed with Zoletil™ (zolazepam30mg/kg; Virbac, Korea). Next, they were anesthetized with sodium pentobarbitone through intraperitoneal injection. Subsequently, a 1.5 cm left upper quadrant incision was made to ligate the left ureter with a silk suture. In the sham group, the left ureter was not ligated; the remaining steps were identical. After sham or UUO operation, treatment drugs were administered continuously at the same dose for 7 days. Mice were all sacrificed at 7 days of UUO/sham operation. After drawing blood from the IVC (inferior vena cava), the mice were sacrificed via cervical dislocation. If anesthesia was compromised during the procedure, an extra 1/2–1/3 dosage of the Rompun/Zoletil mixture was given intraperitoneally, or anesthesia was sustained by inhaling isoflurane for 1–2 seconds. The surgical procedure involved inducing sufficient anesthesia with the agents described above to prevent animal suffering.

## Ethics approval

In all experiments conducted in this paper, the National Research Council and the U.S. National Institutes of Health guidelines for laboratory animal care and use were followed. An Institutional Animal Care and Use Committee (IACUC) of Seoul National University Boramae Medical Center has approved the use of experimental animals at Seoul National University Boramae Medical Center (2022–0023). This study did not include human participants and their biospecimens.

## Establishment of the *in vitro* model

HK-2 cells were incubated and cultured in DMEM/F12 medium. We incubated the HK-2 cells with recombinant TGF-β1 (2 ng/ml; R&D Systems) after serum starvation for 24 h. In addition, HK-2 cells were treated with MN705 at different concentrations (0.05 mg/ml, 0.1 mg/ml, 0.2 mg/ml) simultaneously as rTGF-1 treatment. Following treatments with rTGF-1 and MN705, the cells were harvested after 6-h or 48-h.

Human renal proximal tubule epithelial cells (RPTECs) were purchased from the LONZA (LONZA, CC-2553). RPTECs incubated and cultured in Renal Epithelial cell growth medium (LONZA, CC-3190). We incubated the RPTECs with recombinant TGF-β1 (10 ng/ml; Sigma-Aldrich, T7039). In addition, RPTECs were treated with MN705 at different concentrations (0.05 mg/ml, 0.1 mg/ml, 0.2 mg/ml) simultaneously as rTGF-1 treatment. Following treatments with rTGF-1 and MN705, the cells were harvested after 48-h.

## Histological analyses

In this study, paraffin sections of 4 μm thickness were stained with Masson's trichrome and Sirius red staining agents (all from ScyTek, Logan, Utah, USA). The paraffin-embedded kidneys, which were cut into 4-μm-thick slices, were deparaffinized and hydrated using xylene and ethanol for the immunohistochemical tests. To block the endogenous streptavidin

activity, 3% hydrogen peroxide was applied. After the tissue was deparaffinized, we stained the sections with an anti-collagen 1A1 antibody (Abcam, Cambridge, MA). To detect the primary antibody, a Polkin HRP DAB detection kit (GBI Labs, Bothell, WA, USA) was used. Finally, all sections were counterstained with Mayer's hematoxylin (Sigma-Aldrich). The area of fibrosis and total tissue were measured at ×100 magnification using ImageJ 1.52d software (Wayne Rasband,National Institute ofHealth, U.S.A.).

## Quantitative real-time PCR

We isolated the total RNA from mouse kidney tissues and analyzed its mRNA levels by real-time PCR. We reverse-transcribed 1g of total RNA using oligo(dT) primers and AMV-RT Taq polymerase (Promega, Madison, WI). We performed real-time PCR using either TaqMan probes or the SYBR Green method and primers for α-smooth muscle actin (α-SMA), collagen 1A1, and glyceraldehyde-3-phosphate dehydrogenase (GAPDH). Using the Ct method, relative quantification was performed. Expression levels of mRNA were normalized to those of GAPDH mRNA.

## Western blot analysis

The kidney tissues and cells were harvested and collected. Proteins were extracted and separated in 10% sodium dodecyl sulfate-polyacrylamide gels before being transferred to Immobilon-FL polyvinylidene difluoride membranes (Millipore, Bedford, MA). After the nonspecific proteins had been blocked, the membranes were incubated with specific primary antibodies overnight at 4°C, before the incubation with specific secondary antibodies. The secondary antibodies used were anti-rabbit IgG or anti-mouse IgG antibodies (all from Cell Signaling Technology, Danvers, MA, USA).

## Statistical analyses

All cell culture-related experiments were analyzed using one-way analysis of variance (ANOVA) and Tukey's post-hoc tests to determine significant differences among treatments. P values <0.05 was statistically significant. Statistical analyses were performed using IBM SPSS 20.0 and GraphPad Prism 8.0 (Graph-Pad Software, San Diego, CA).

## Results

### Screening and selection of *Rubus coreanus* as potential substance

The process of screening and selection of extract of *Rubus coreanus* (MN705, lot number MVW021105, Medvill Co., Ltd. Korea) among 9 natural product extracts is shown in Fig 1A. The cell viability was measured by MTT assay. A number of natural product extracts showed cytotoxicity with decreased cell viability over a range of concentrations. MN705 had a relatively enhanced effect on cell viability at the concentration of 0.5 mg/ml, and had little damage to cell viability at a higher concentration except 10 mg/ml dose (Fig 1B). The visual appearance and color of natural products dissolved in distilled water is shown in Fig 1C. Through this process, MN705 was ultimately selected as the final natural substance candidate. MN705 is a compound extracted from *Rubus coreanus* and includes various bioactive compounds, such as anthocyanins, tannins, gallic acid, and ellagic acid.

### The anti-fibrotic effects of MN705 on HK-cells

Cell morphology showed that HK-2 cells lost their epithelial appearance and presented elongated and spindle-shaped morphology after being stimulated with 2 ng/ml TGF-β for 48-h (Fig 2A). MN705 (0.2 mg/ml) considerably reduced spindle-like morphology induced by TGF-β stimulation and decreased the expression of fibronectin. In western blot, MN705 administration increased the antioxidative enzyme of MnSOD (0.76 ± 0.03, 1.42 ± 0.12, 1.48 ± 0.24, and 1.61 ± 0.13 fold changes compared to control, overall $P < 0.001$; TGF-β 2 ng/ml with vehicle, MN705 0.05 mg/ml, 0.01 mg/ml, and 0.02 mg/ml, respectively) and inhibited the fibrosis-related markers of fibronectin (4.58 ± 0.41, 3.76 ± 0.37, 3.11 ± 0.35, and

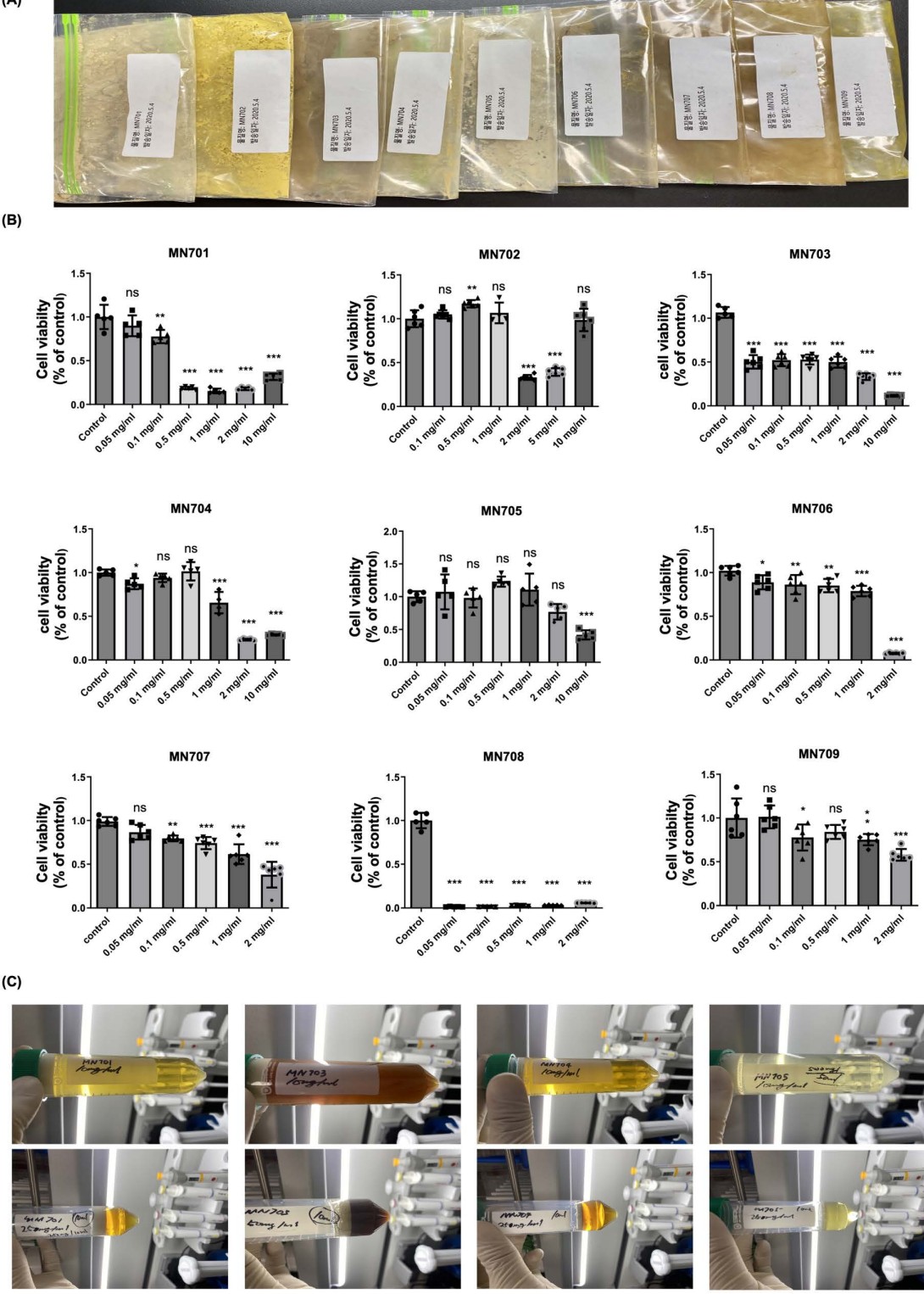

**Fig 1. The screening process of MN705.** (A) Water-soluble powder extract of natural extracts of MN701 to MN709. (B) Measurement of cell viability. Assessment of cell activity by MTT assay. The data are presented as the mean ± SD of per group. (C) The color of their water-soluble substances. *P < 0.05, ** P < 0.01, ***P < 0.001.

**(A)**

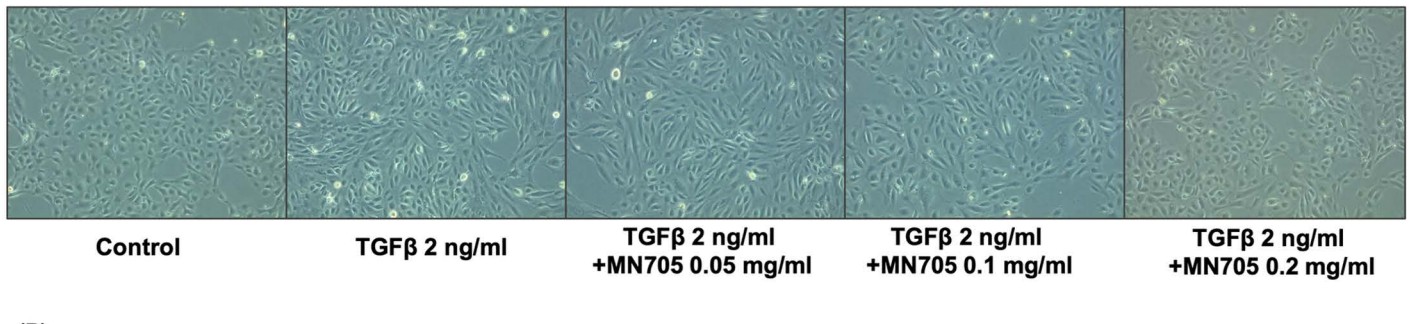

| Control | TGFβ 2 ng/ml | TGFβ 2 ng/ml +MN705 0.05 mg/ml | TGFβ 2 ng/ml +MN705 0.1 mg/ml | TGFβ 2 ng/ml +MN705 0.2 mg/ml |

**(B)**

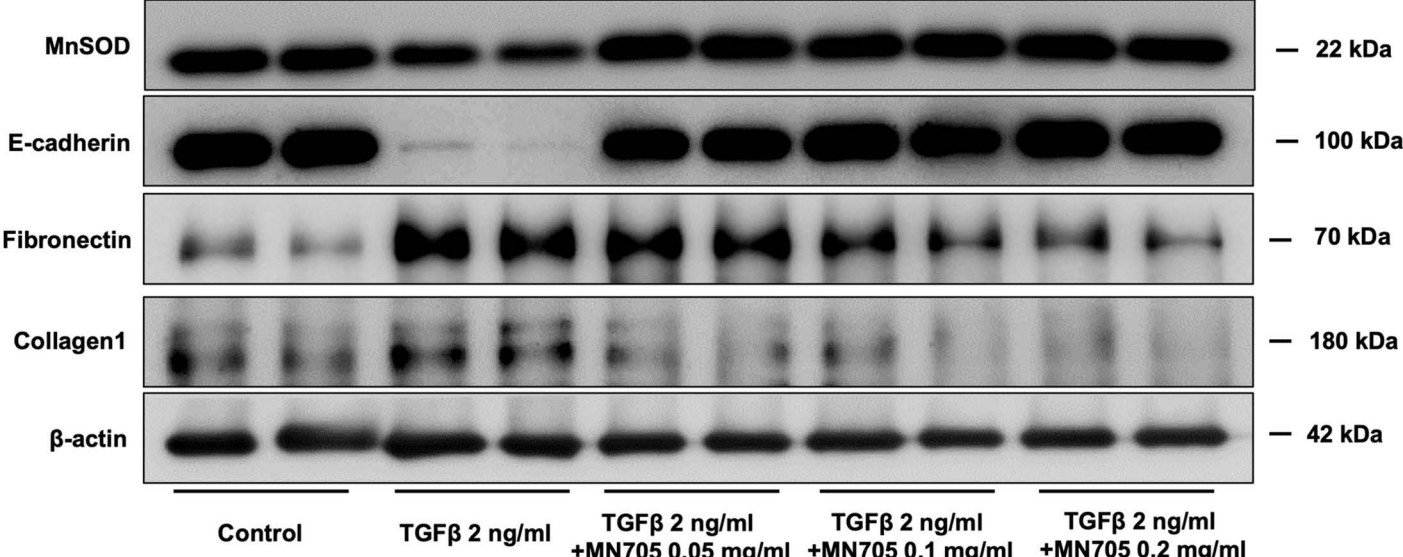

**(C)**

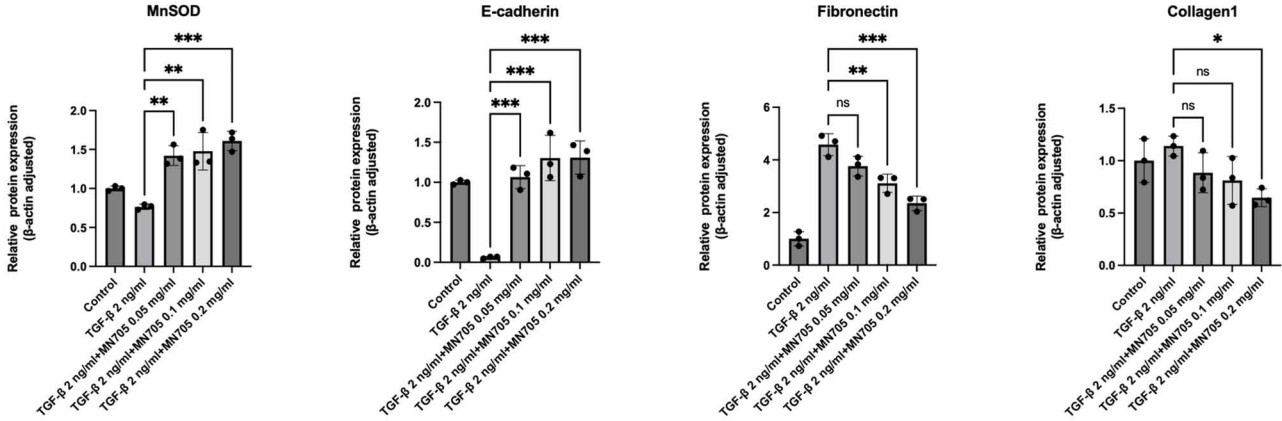

**Fig 2. Changes associated with kidney fibrosis in TGF- β1-induced fibrosis responses of HK-2 cell lines.** (A) The morphological changes of cells in each group were observed by light microscope (×200). (B) Protein expression of MnSOD, E-cadherin, fibronectin, and collagen-1. A representative band is shown for each protein. (C) Measurement of western blot results. The data are presented as the mean±SD of per group. $^*P<0.05$, $^{**}P<0.01$, $^{***}P<0.001$.

2.35±0.27 fold changes compared to control, overall $P<0.001$; TGF-β 2 ng/ml with vehicle, MN705 0.05 mg/ml, 0.01 mg/ml, and 0.02 mg/ml, respectively) and collagen-1 with increased expression of E-cadherin ([Fig 2B], [2C]).

To explore the potential mechanism of anti-fibrosis through MN705, we examined whether MN705 inhibited the activation of TGF-β -induced Smad pathway *in vitro* HK-2 cell models. MN705 down-regulates TGF-β/Smad signaling pathway in HK-2 cells ([Fig 3]). [Fig 3A] and [3B] show the western blot results of TGF-β 6-h challenged model and [Fig 3C] and [3D] are the results of TGF-β 48-h challenged model. TGF-β treatment for 6-h increased the expression of phospho-Smad2/3 and Smad ubiquitination regulator 1 (Smurf1), which was significantly reduced by MN705 treatment ([Fig 3A], [B]). In the TGF-β-treated (48-h) HK-2 cell *in vitro* model, Smurf1 expression increased and significantly attenuated by MN705 treatment (1.70±0.35, 1.48±0.24, 0.65±0.24, and 0.18±0.19 fold changes compared to control, overall $P<0.001$; TGF-β 2 ng/ml with vehicle, MN705 0.05 mg/ml, 0.01 mg/ml, and 0.02 mg/ml, respectively); however, phospho-Smad2/3 decreased with prolonged exposure to TGF-β and the decrease after MN705 treatment was not significant ([Fig 3C], [D]). Therefore, we speculate that MN705 alleviates fibrosis through the TGF-β/Smad pathway in the early stage of fibrosis.

The results found in the HK-2 cell line were validated using human renal proximal tubular epithelial cells. After MN705 treatment, the epithelial cell morphology altered by TGF-β 2 ng/ml for 48-h was restored to a more elongated and spindle-shaped morphology ([Fig 4A]). Western blots were employed to investigate alterations of proteins associated with anti-oxidation and kidney fibrosis. MN705 led to enhanced expression of the antioxidant MnSOD, reduced fibronectin levels, and diminished phospho-Smad2/3 ([Fig 4B], [4C]).

## The anti-fibrotic effects of MN705 on UUO mice

[Fig 5] presents the histopathological change in the sham and UUO mice treated with MN705. UUO mice kidney showed loss of kidney tubular epithelial cells, interstitial inflammation infiltration, and collagen deposition. These abnormalities were attenuated when 300 and 600 mg/kg/d of MN705 were administered to the UUO mice ([Fig 5A] and [5B]). In Masson's trichrome staining, MN705 treatment decreased the area of fibrosis significantly (17.2±2.3%, 12.4±1.7%, and 10.6±1.3%, overall $P<0.001$; UUO with vehicle, MN705 300 mg/kg/day, and MN705 600 mg/kg/day, respectively). In immunohistochemical staining of collagen-1, the kidney tissue areas of collagen-1 decreased significantly after MN705 treatment (28.3±3.5%, 20.7±3.2%, and 15.5±3.9%, overall $P<0.001$; UUO with vehicle, MN705 300 mg/kg/day, and MN705 600 mg/kg/day, respectively).

UUO mice showed typical kidney fibrosis changes characterized by elevated protein expression of the α-SMA (19.8±3.1 fold changes compared to sham group, $P<0.001$) and collagen type I (18.0±10.8 fold changes compared to sham group, $P<0.001$) compared with that in sham mice ([Fig 6]). MN705 inhibited successfully kidney fibrosis in the UUO model. Western blotting showed that MN705 markedly reversed the elevation of α-SMA [19.8±3.1, 17.7±6.1 ($P=0.851$ vs. UUO group), and 12.9±5.9 ($P=0.046$ vs. UUO group) fold changes, overall $P<0.001$; UUO with vehicle, MN705 300 mg/kg/day, and MN705 600 mg/kg/day, respectively] protein levels; although, the expression of collagen type I did not significantly decreased ([Fig 6A], [6B]). Furthermore, MN705 treatment substantially down-regulated the mRNA expression level of α-SMA [12.4±8.1, 9.1±1.8 ($P=0.116$ vs. UUO group), and 7.5±1.7 ($P=0.028$ vs. UUO group) fold changes, overall $P<0.001$; UUO with vehicle, MN705 300 mg/kg/day, and MN705 600 mg/kg/day, respectively], collagen-1 [36.7±24.3, 25.6±14.5 ($P=0.529$ vs. UUO group), and 13.6±10.9 ($P=0.04$ vs. UUO group) fold changes, overall $P<0.001$; UUO with vehicle, MN705 300 mg/kg/day, and MN705 600 mg/kg/day, respectively] induced by UUO operation ([Fig 6C]).

MN705 inhibits kidney fibrosis through TGF-β/Smad pathway, which is consistent with the results of cell experiments. In western blot, MN705 significantly reduces phospho-Smad2/3 and total Smad2/3 protein abundance ([Fig 6A], [6B]). In addition, MN705 significantly reduces TGF-β and Smurf1 protein abundance, and the expression of Smad7 [21.4±4.4, 27.0±6.4 ($P=0.074$ vs. UUO group), and 33.6±2.27 ($P<0.001$ vs. UUO group) fold changes, overall $P<0.001$; UUO with vehicle, MN705 300 mg/kg/day, and MN705 600 mg/kg/day, respectively] was increased after MN705 treatment (S1A,B Fig). These results show that MN705 can effectively inhibit kidney fibrosis via TGF-β/Smad pathway modulation.

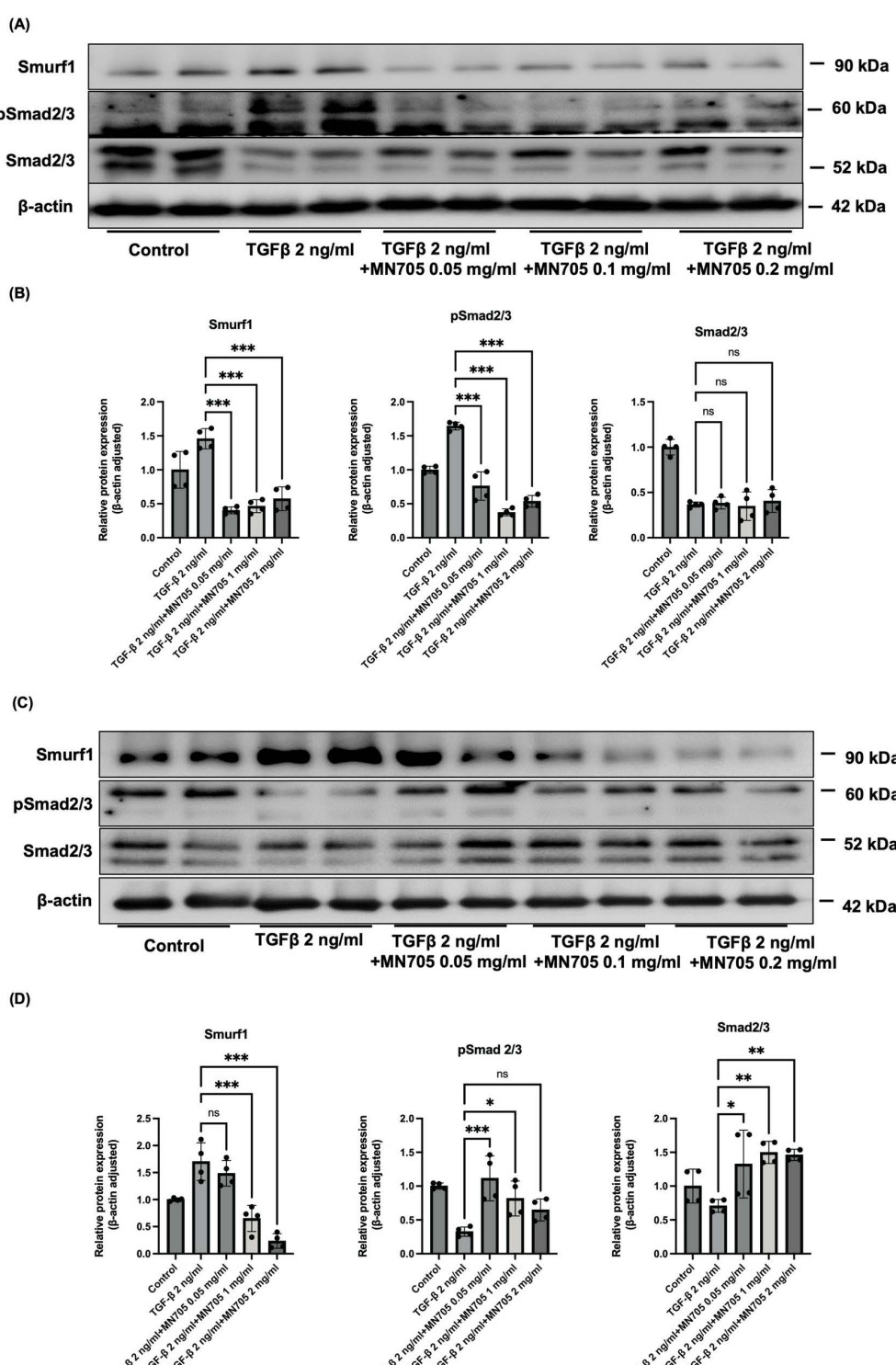

**Fig 3. Effect of MN705 treatment on TGF- β1/Smad pathway in TGF-β1-induced fibrosis responses of HK-2 cell lines.** (A) Protein expression of Smurf1, phospho-Smad2/3, Smad2/3. HK-2 cells were treated with TGF-β1 for 6-h. A representative band is shown for each protein from triplicate experiments performed under the same experimental design. (B) Measurement of western blot results. (C) Protein expression of Smurf1, phospho-Smad2/3, Smad2/3. HK-2 cells were treated with TGF-β1 for 48-h. (D) Measurement of western blot results. The data are presented as the mean ± SD of per group. $^*P<0.05$, $^{**}P<0.01$, $^{***}P<0.001$.

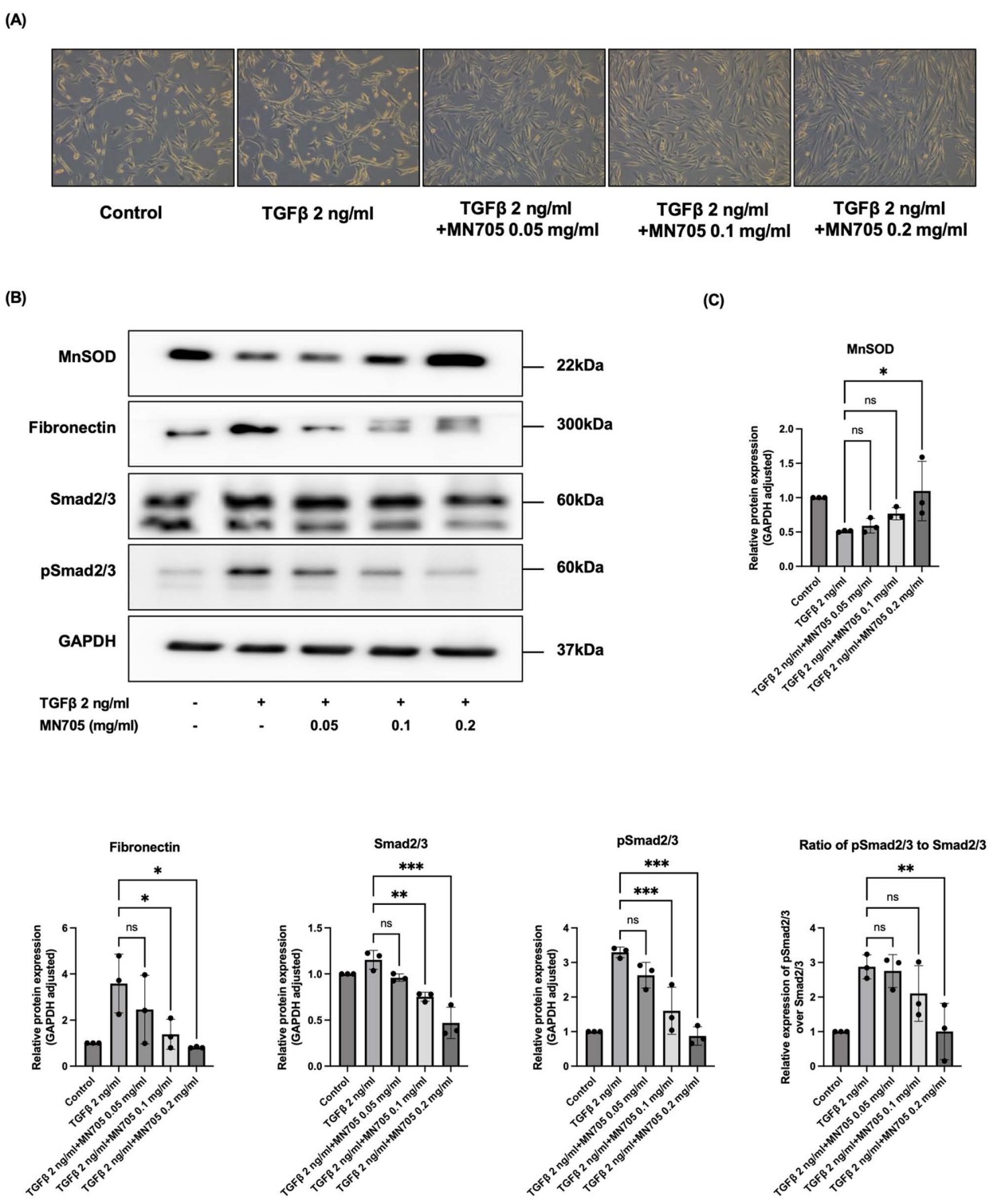

**Fig 4. Validation of MN705 treatment effects in human proximal tubular epithelial cells during TGF- β1-induced fibrosis responses.** (A) The morphological changes of cells in each group were observed by light microscope (×100). (B) Protein expression of MnSOD, fibronectin, Smad2/3, and phospho-Smad2/3. A representative band is shown for each protein. (C) Measurement of western blot results. The data are presented as the mean±SD of per group. $^*P<0.05$, $^{**}P<0.01$, $^{***}P<0.001$.

**(A)**

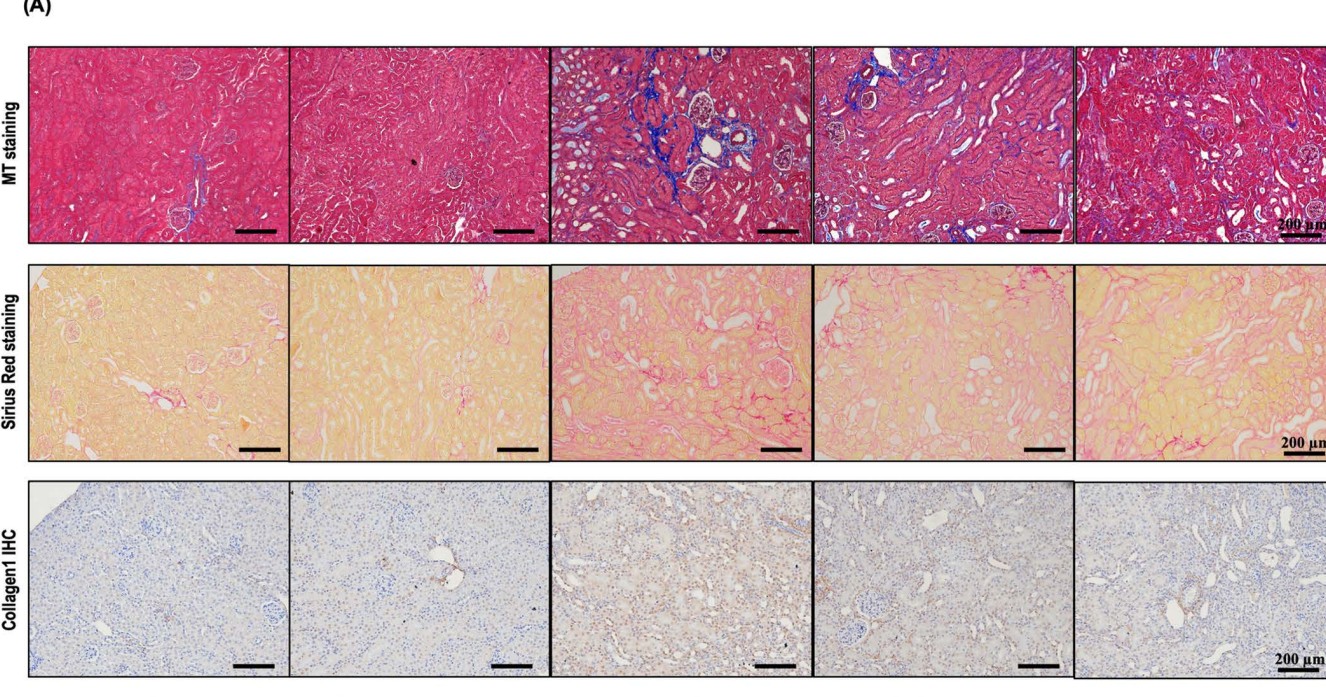

Sham+DW          Sham+MN705 600 mg          UUO+DW          UUO+MN705 300 mg          UUO+MN705 600 mg

**(B)**

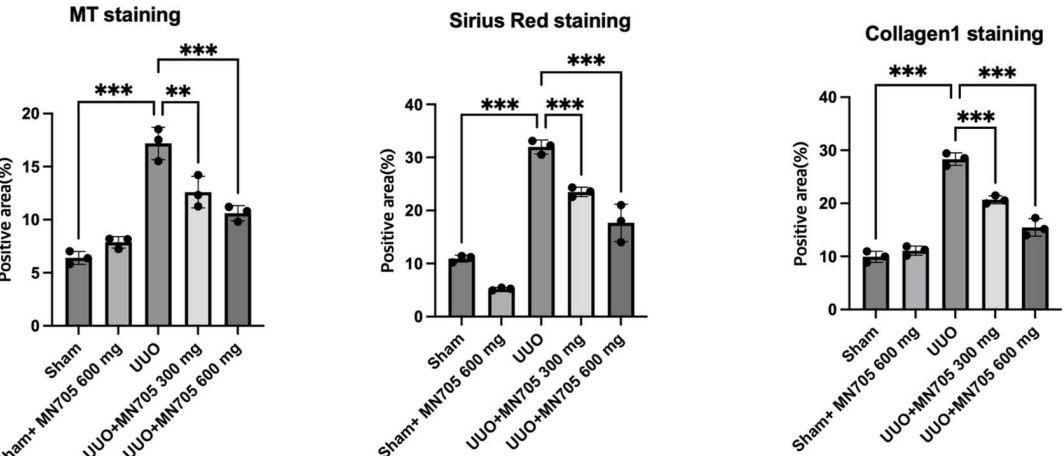

**Fig 5. Changes related to kidney staining in UUO mouse model.** (A) Interstitial fibrosis displayed by Masson's Trichrome or Sirius Red staining and immunohistochemical staining for Col1a1 proteins (×200 magnification). (B) Measurement of interstitial fibrosis area and tissue expression areas of Col1a1. The data are presented as the mean±SD of per group. $^*P<0.05$, $^{**}P<0.01$, $^{***}P<0.001$.

In addition, we also identified changes in anti-oxidative enzymes and oxidative stress in an *in vivo* model. The expression of MnSOD decreased significantly after the UUO operation (Fig 7A). In the MN705 treatment group, reduced MnSOD expression showed a relatively increased response with no statistical significance (Fig 7C). We measured the level of

**(A)**

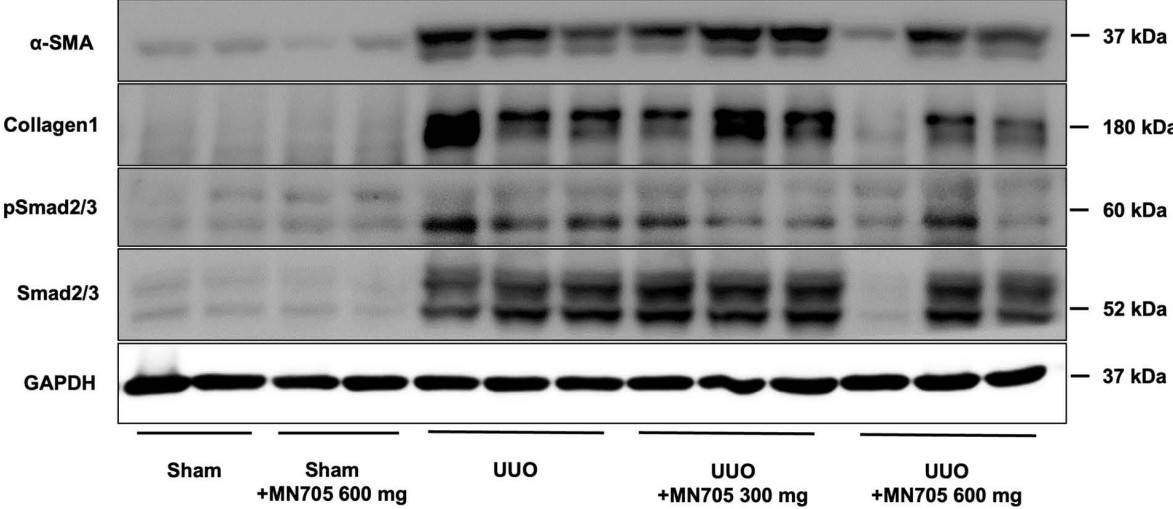

**(B)**

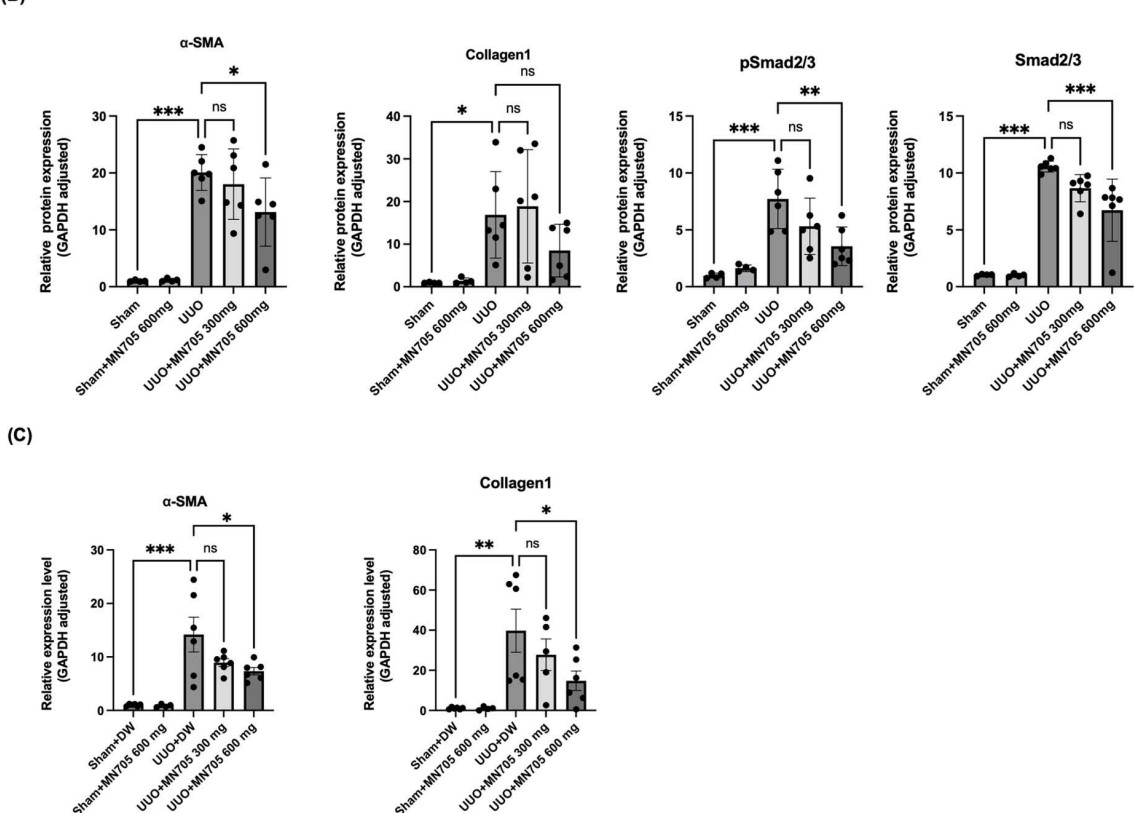

**(C)**

**Fig 6. Changes associated with kidney fibrosis and effects of MN705 treatment on TGF- β1/Smad pathway in mouse model of UUO.** (A) Protein expression of α-SMA, collagen-1, phospho-Smad2/3, and Smad2/3. (B) Measurement of western blot results. The data are presented as the mean±SD of per group. (C) Expression of α-SMA and collagen-1 mRNA in control and experimental mouse extracts by RT-PCR. Measurement of RT-PCR results. The data are presented as the mean±SD of per group. *$P<0.05$, ** $P<0.01$, ***$P<0.001$.

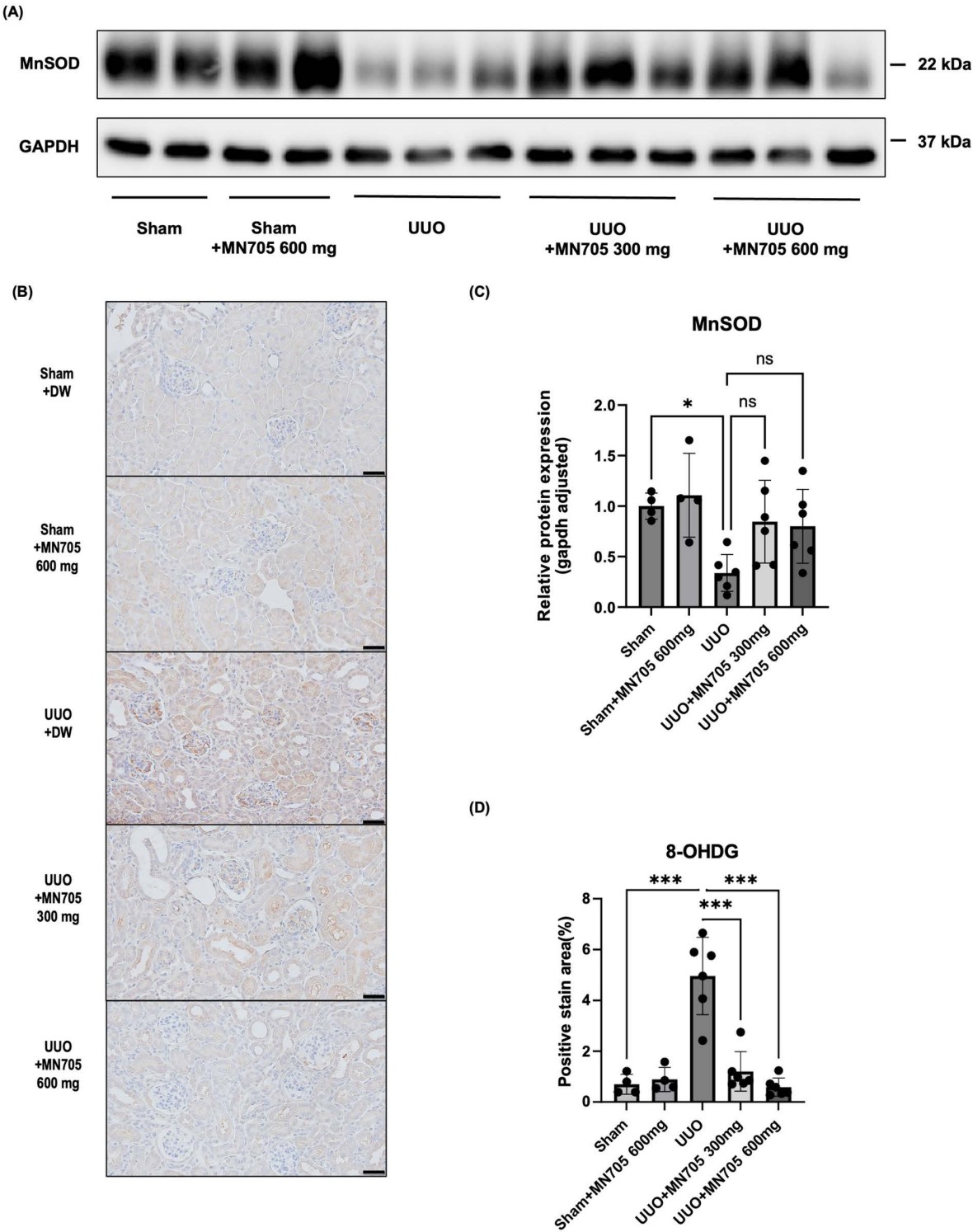

**Fig 7. Changes of anti-oxidative enzyme and oxidative stress after in a mouse model of kidney fibrosis.** (A) Protein expression of MnSOD. (B) Immunohistochemical staining for 8-OHDG (×400 magnification, scale bar 20 μm). (C) Measurement of western blot results and (D) areas of immunohistochemical stain. The data are presented as the mean±SD of per group. *P<0.05, ** P<0.01, ***P<0.001.

oxidative stress using 8-OHDG immunohistochemical staining. The UUO group showed significantly increased tissue expression levels of 8-OHDG (Fig 7B, 7D). After MN705 treatment, the MN705 treatment group showed a significant decrease in oxidative stress expressed with 8-OHDG.

## Discussion

*Rubus coreanus* is a wild berry belonging to *Rosaceae* genus of which application areas have been expanded due to many health effects including anti-oxidation. We aimed to investigate whether the treatment of a water-soluble extract of *Rubus coreanus* (MN705) can attenuate kidney fibrosis in unilateral ureteral obstruction (UUO) mice kidney fibrosis and *in vitro* TGF-β challenged HK-2 cell models. In the histopathological specimen of Masson's trichrome stain, areas of kidney interstitial fibrosis attenuated in the treatment group. In the western-blot analysis and RT-PCR, the protein abundance and mRNA expression of α-SMA decreased significantly in the treatment group. The anti-fibrotic properties of MN705 were strongly linked to its ability to regulate the signaling of the TGF-beta/Smad pathway and reduce oxidative stress. *In vitro* experiments, HK-2 cells treated with TGF-β and MN705 decreased significantly protein expression of fibronectin in a dose-dependent manner with increase of the antioxidative enzyme of MnSOD. Smurf1 and phospho-Smad2/3 also decreased in a dose-dependent manner especially in the early phase. MN705 may alleviate kidney fibrosis through Smurf1 and Smad pathways.

   *Rubus coreanus* extract has been found to have multiple biological effects in many *in vitro* experiments and *in vivo* animal studies. A study showed that *Rubus coreanus* extract showed high cytotoxicity to sensitive leukemia HL60 cell line and its MDR subline, and the resistance factor (RF) value of the extract was low, in the range of 0.32–2.0 [10]. The inhibitory effect of *Rubus coreanus* on the production of inflammatory mediators varies with the ripening stage of fruit. Yang et al. found that *Rubus coreanus* had the strongest anti-inflammatory effect in the immature stage. It inhibited the production of inflammatory mediators such as NO, PGE2, and other inflammatory factors [11]. Hong et al. evaluated the protective effect of *Rubus coreanus* on acute liver injury induced by carbon tetrachloride in mice. Their study confirmed that hepatoprotective effects of Rubus species were derived from the inhibition of hepatic production of MDA and NO, and intensification of SOD activity [12]. Wu et al. reported that *Rubus coreanus* reduced liver fibrosis by inducing apoptosis and trans-differentiation of activated hepatic stellate cells [13]. Several studies have suggested that *Rubus coreanus* may have anti-fibrotic effects in different organs. In a rat model of liver fibrosis induced by carbon tetrachloride, administration of *Rubus coreanus* extract significantly reduced collagen deposition and improved liver function markers [14]. In addition, *Rubus coreanus* extract attenuated atherosclerosis in a mouse model of pulmonary fibrosis induced by bleomycin [15].

   *Rubus coreanus* has been used as a Chinese herbal medicine for years, it has been widely used to strengthen kidney function [16]. In AKI model induced by cisplatin, *Rubus coreanus* extract reduced cisplatin-induced acute tubular necrosis and histological changes associated with the attenuation of oxidative stress and the preservation of antioxidant enzymes [17]. *In vitro* studies have also demonstrated the protective effects of *Rubus coreanus* extract on kidney cells. *Rubus coreanus* extract was found to reduce oxidative stress and inflammation in cultured kidney cells exposed to various insults, such as high glucose or cisplatin treatment [18]. The anti-fibrotic effects of *Rubus coreanus* may be mediated by the suppression of oxidative stress and inflammation, which are known to promote fibrosis [19]. The constituents of *Rubus coreanus* can vary, but it generally contains a range of bioactive compounds that contribute to its potential health benefits. The representative components of *Rubus coreanus* include polyphenols, vitamins, and minerals. Polyphenolic compounds, such as anthocyanins, flavonoids, and ellagic acid have been widely studied for their antioxidant properties [20]. *Rubus coreanus* may contain various vitamins and minerals, such as vitamin C and manganese, which also contribute to its antioxidant capacity. The antioxidant properties of *Rubus coreanus* can play a crucial role in neutralizing free radicals and reducing the production of reactive oxygen species and pro-inflammatory cytokines, thereby inhibiting fibrogenesis [21]. There has been research investigating the potential role of antioxidants in treating kidney fibrosis in experimental studies. Antioxidants were studied for their ability to reduce oxidative stress, inflammation, and cellular damage, which

were associated with the development of fibrosis. N-Acetylcysteine (NAC) is a precursor to glutathione, a powerful endogenous antioxidant. Some studies have investigated the potential of NAC and it's anti-oxidative effect in ameliorating kidney fibrosis [22]. Resveratrol is a polyphenol found in certain foods, including red grapes and berries. Chowdhury et al. demonstrated that resveratrol treatment reduced kidney fibrosis in a high-fat diet rat model, possibly through its antioxidant and anti-inflammatory effects [23]. However, the detailed mechanism of anti-fibrosis has not been fully investigated.

Signaling pathways involved in fibrogenesis can be a good target for kidney anti-fibrosis. *Rubus coreanus* extract has been shown to inhibit the TGF-β signaling pathway [24]. After kidney damage, TGF-β is released from the injured tubular epithelial cell and triggers kidney fibroblasts [25]. Activated fibroblast by TGF-β consequently produces collagen and other extracellular matrix proteins. TGF-β can induce kidney fibrosis by activating classical (Smad-based) signaling pathways, resulting in stimulation of myofibroblasts, increased extracellular matrix (ECM) synthesis, and inhibition of ECM breakdown [26]. Smad proteins are intracellular mediators that transduce the signals of TGF-β from the cell surface to the nucleus [27]. Moreover, Smad2/3 is the essential mediator of TGF signal transduction and is harmful to kidney inflammation and fibrosis [28].

TGF-β/Smad was an independent and excessively activated pathway that downstream pro-fibrotic genes expression including fibronectin and α-SMA contributed to profibrotic gene transcription and collagen deposition in kidney cells [29]. Smad3 overexpression leads to kidney fibrosis through epithelial-mesenchymal transformation, monocyte influx, and collagen deposition. Moreover, Smad3 targeted therapy protects against kidney injury, programmed cell death, and inflammation through NOX4-dependent oxidative stress [30]. Under disease conditions, Smad3 strongly reacts, inducing Smurf1, its physical interaction with Smad7, and causing ubiquitination dependent degradation of the kidney Smad7 protein [28]. Song et al. found that Smurf1 inhibition by siRNA led to elevated Smad7 expression and prevention of kidney fibrosis [31]. Smad7 binds to Smurf2 to form E3 ubiquitin ligase, targeting TGF-β receptor degradation [32].

The response of TGF-β/Smad signaling and phospho-Smad2/3 can differ between the early and late stages of fibrosis. In the early stages of fibrosis, the TGF-β/Smad signaling pathway promotes the activation of fibroblasts, leading to the production and deposition of extracellular matrix components. In late fibrosis, there is often sustained and prolonged activation of TGF-β signaling. This persistent activation contributes to the chronicity of the fibrotic process. In animal studies, UUO causes continuous pressure damage to the kidneys, accelerating fibrosis. Several *in vitro* studies have shown that activation of phospho-Smad2 occurs between 45 minutes and 1 hour after stimulation and disappears after 4–5 hours [33,34]. In this study, stimulation of HK-2 cells with TGF-β resulted in a decrease in Smad2/3 activation at 48 hours, but we observed a significant increase in the 6-hour early model. Therefore, to observe Smad activation in cellular models, it is necessary to use an early model within 6 hours. The understanding of these differences is crucial for developing *in vitro* model and targeted therapeutic strategies to intervene in the fibrotic process at different stages.

Although the direct mechanism of regulation of TGF-β/Smad signaling by antioxidants was not unveiled in this study, treatment with *Rubus coreanus* effectively regulated TGF-β/Smad signaling in both *in vitro* and *in vivo* models and alleviated kidney fibrosis. There has been some evidence that support anti-oxidation via phenolic compounds of phytochemicals can attenuate kidney fibrosis via Smad pathway. Berberine (isoquinoline alkaloid; found in Coptidis Rhizoma and Cortex Phellodendri) had anti-fibrosis effects via inhibition of Smad2/3 in streptozocin-induced diabetic kidney disease model [35]. Dioscin (steroid saponin; found in Dioscoreae rhizome) showed anti-oxidative and anti-fibrosis effects with inhibition of Smad3 and activation of Smad7 in 10% fructose-fed mice model [36]. Sinomenine (Alkaloid; found in Sinomenium acutum) also had anti-oxidative and anti-fibrosis effects with inhibition of Smad3 in UUO mice model and TGF-β/$H_2O_2$-induced *in vitro* fibrosis model [37]. This study has the unique advantage of being the first to show the antifibrotic properties of *Rubus coreanus* in kidney fibrosis. Additionally, it proposes that the modulation of TGF-β/Smad signaling, along with the anti-oxidative effects, is the underlying mechanism of anti-fibrosis. In addition, the effects of *Rubus coreanus* on TGF-β/Smad signaling, antioxidants, and preventing fibrosis were confirmed in both animal study and two *in vitro* models. While these studies suggest a potential therapeutic role for antioxidants in kidney fibrosis, it's essential to approach the findings with caution. The effectiveness of antioxidants may vary depending on the specific conditions and causes of kidney fibrosis. The UUO model induces

kidney fibrosis through a sustained pressure effect within the kidney. This pathophysiological trait contrasts with most clinical disorders associated with kidney fibrosis. In addition, TGF-β is a principal pro-fibrotic factor that activates myofibroblasts and serves as a crucial mediator in the progression of kidney fibrosis. However, it is essential to experimentally replicate many circumstances associated with kidney fibrosis, including hypoxia, infection, and inflammation, to facilitate the generalization of experimental results. In addition, this study only confirmed the preventive effect of *Rubus coreanus* on anti-fibrosis. Further experiments and clinical studies are necessary to determine if the *Rubus coreanus* has a therapeutic effect on kidney fibrosis. At last, the anti-fibrotic impact of *Rubus coreanus* was only observed in the 7-day model of the UUO mouse. Therefore, it is worthwhile to investigate whether this anti-fibrotic effect persists over shorter or longer time intervals.

In conclusion, *Rubus coreanus* attenuates kidney fibrosis *in vivo* mice UUO kidney fibrosis model and *in vitro* TGF-β challenged HK-2 cell models. Treatment with *Rubus coreanus* increased the activity of antioxidative enzyme MnSOD. Anti-fibrotic and anti-oxidative effects of *Rubus coreanus* are related with suppression of TGF-β/Smad pathway, inhibition of Smurf1 and phospho-Smad2/3 and activation of Smad7. The antioxidant and antifibrotic effects of *Rubus coreanus* through inhibition of TGF-β/Smad pathway may be useful for the development of therapeutics for kidney fibrosis in the future. Further studies in other kidney disease models and clinical trials are needed to investigate whether these beneficial effects of *Rubus coreanus* can improve the prognosis of kidney disease.

## Supporting information

**S1 Fig. Effect of MN705 treatment on TGF-β1/Smad pathway in a mouse model of kidney fibrosis.** (A) Protein expression of TGF-β, Smurf1, and Smad7. (B) Measurement of western blot results. The data are presented as the mean±SD of per group. $^{*}P < 0.05$, $^{**}P < 0.01$, $^{***}P < 0.001$.
(PDF)

**S1 File. Western blot band raw data images.**
(PDF)

## Acknowledgments

We appreciate the technical support from SMG-SNU Boramae Medical Center. There is nothing to disclose.

## Author contributions

**Conceptualization:** Wencheng Jin, Jeonghwan Lee.

**Data curation:** Ara Cho, Bogyeong Cho, Nayeon Shin, Jeonghwan Lee.

**Formal analysis:** Wencheng Jin, Ara Cho, Bogyeong Cho, Nayeon Shin, Jeonghwan Lee.

**Funding acquisition:** Jeonghwan Lee.

**Investigation:** Wencheng Jin, Ara Cho, Bogyeong Cho, Jung Pyo Lee, Jeonghwan Lee.

**Methodology:** Wencheng Jin, Ara Cho, Nayeon Shin, Jung Pyo Lee, Jeonghwan Lee.

**Resources:** Jung Pyo Lee.

**Software:** Wencheng Jin, Ara Cho, Jeonghwan Lee.

**Supervision:** Yun Kyu Oh, Chun Soo Lim, Jung Pyo Lee, Jeonghwan Lee.

**Visualization:** Bogyeong Cho, Jeonghwan Lee.

**Writing – original draft:** Wencheng Jin.

**Writing – review & editing:** Wencheng Jin, Bogyeong Cho, Jeonghwan Lee.

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
