## [Decision Letter · Decision Letter 0]

31 Aug 2023

PONE-D-23-21717Rubus coreanus extract attenuates kidney fibrosis through TGF-β/Smad pathway inhibitionPLOS ONE

Dear Dr. Lee,

Thank you for submitting your manuscript to PLOS ONE. After careful consideration, we feel that it has merit but does not fully meet PLOS ONE’s publication criteria as it currently stands. Therefore, we invite you to submit a revised version of the manuscript that addresses the points raised during the review process.

Your manuscript has been reviewed by experts in the field. Their comments are attached for your view. ==============================

We look forward to receiving your revised manuscript.

Kind regards,

Qiang Ding, Ph.D.

Academic Editor

PLOS ONE

3. To comply with PLOS ONE submissions requirements, in your Methods section, please provide additional information regarding the experiments involving animals and ensure you have included details on (1) methods of sacrifice, (2) methods of anesthesia and/or analgesia, and (3) efforts to alleviate suffering

4. We suggest you thoroughly copyedit your manuscript for language usage, spelling, and grammar. If you do not know anyone who can help you do this, you may wish to consider employing a professional scientific editing service.

The name of the colleague or the details of the professional service that edited your manuscript.

Reviewers' comments:

Reviewer's Responses to Questions

**Comments to the Author**

1. Is the manuscript technically sound, and do the data support the conclusions?

Reviewer #1: Yes

Reviewer #2: No

2. Has the statistical analysis been performed appropriately and rigorously? 

Reviewer #1: Yes

Reviewer #2: No

3. Have the authors made all data underlying the findings in their manuscript fully available?

Reviewer #1: Yes

Reviewer #2: Yes

4. Is the manuscript presented in an intelligible fashion and written in standard English?

Reviewer #1: Yes

Reviewer #2: Yes

5. Review Comments to the Author

Reviewer #1: The manuscript described potential protective effects of Rubus coreanus extract (RCEs) on kidney fibrosis through TGF-b/Smad pathway inhibition. They characterized dose-dependent effects of RCEs on TGF-b-stimulated fibrotic signaling in HK-2 cells in vitro and also in vivo animal model of kidney fibrosis. The strength of the manuscript is the identification on the impact of RCEs, that suppresses TGF-b/Smad2/Smurf profibrotic signaling pathway while induction of anti-fibrotic signaling of Smad7 and anti- oxidant MnSOD, that resulted into reduced fibrotic tissue response in UUO animal model of renal fibrosis. Overall, it is descriptive and there are several weaknesses that need to be further clarified. Followings are the specific comments:

1. In Figure 3, significantly different regulations of TGF-b-stimulated Smad signaling in HK-2 cells between acute and chronic exposures were noted. It may need additional discussion to explain why this happen and the impact of clinical application of RCEs.

2. Quantitative Western blot evaluations on TGF-b-stimulated Smad2/3 were not well established based on the band pattern, that need to be consistently applied for accurate quantitative measurement.

3. RCEs are very complex natural product. It may help to use specific component of the complex to characterize in vivo and in vitro effect. Al least some discussions on this aspect.

4. They provide only preventive effect, testing therapeutic effect would be more promising for clinical application.

Reviewer #2: while the paper presents an interesting research for kidney fibrosis, there are significant questions to be addressed before the conclusion and mechanism can be appropriately supported.

1. What is the rationale for using TGFb in cells, while the in vivo model using Rubus coreanus which has anti-oxidative effects?

2. There is a disconnection between the mechanism investigated and results. Results are mainly fibrosis, nothing to do with anti-oxidation effects in cells and animal UUO models.

3. cell results and animal results are not comparable.

4. although there is a dose response, not sure how the doses were selected, or any unwanted effects. What will be lowest dose to be effective?

5. what about time responses in animal models?

6. Image quality needs to be improved.

7. Mechanism involved is not clear, or at least not supported by the current data for the effect of Rubus coreanus

6. PLOS authors have the option to publish the peer review history of their article (what does this mean? ). If published, this will include your full peer review and any attached files.

**Do you want your identity to be public for this peer review?** For information about this choice, including consent withdrawal, please see our Privacy Policy .

Reviewer #1: No

Reviewer #2: No

---

## [Author Response · Author response to Decision Letter 1]

26 Mar 2024

The authors appreciate the decision and comments from the editor and reviewers. important comment. The authors have made the necessary improvements. In this revision, we have further confirmed the extent of oxidative damage (8-OHDG) and changes in antioxidant markers (MnSOD) in animal experiments. Furthermore, the authors have rectified the western blot measurement error and enhanced the image quality to ensure clear visibility of all numbers and letters when zoomed in. Thank you again for the time and opportunity to conduct further experimentation for this revision. Through the revision process, the authors were able to fill in the gaps and provide a more logical explanation for our findings.

The required files of a rebuttal letter (file labeled 'Response to Reviewers'), a marked-up copy of the manuscript (file labeled 'Revised Manuscript with Track Changes'), and an unmarked version of the revised paper without tracked changes ('Manuscript') were all prepared and uploaded. Financial disclosures are newly added to the editorial manager system. The authors have endeavored to follow the guidelines for resubmitting figure files. The authors ensure that this manuscript meets PLOS ONE's style requirements. The authors also attached the original uncropped and unadjusted images underlying all blot or gel results. In the cover letter, the authors noted that blot/gel image data are in Supporting Information. In the Methods section, we provided additional information regarding the experiments involving animals and ensured the authors included details on (1) methods of sacrifice, (2) methods of anesthesia and/or analgesia, and (3) efforts to alleviate suffering. The authors have used a professional scientific editing service to ensure that the English language is correct. The authors moved the ethics declaration, which appeared after the main text, to the methods section.

With this revision, we hope to have this study published in PLOS ONE. Thank you very much.

1-1. Thank you for the important comment. TGF-β signaling plays a crucial role in the process of fibrosis, a condition characterized by the excessive accumulation of extracellular matrix components in tissues. The canonical TGF-beta signaling pathway involves the activation of Smad proteins, particularly Smad2/3. The response of TGF-β/Smad signaling and phospho-Smad2/3 can differ between the early and late stages of fibrosis.

In response to tissue injury or inflammation, TGF-β is released and activated. The activated TGF-β binds to its receptors on the cell surface, leading to the phosphorylation of Smad2 and Smad3. Phosphorylated Smad2 and Smad3 form a complex with Smad4. This complex translocates into the nucleus, where it regulates the transcription of target genes involved in fibrosis, such as those encoding extracellular matrix proteins. In the early stages of fibrosis, the TGF-beta/Smad signaling pathway promotes the activation of fibroblasts, leading to the production and deposition of extracellular matrix components.

In late fibrosis, there is often sustained and prolonged activation of TGF-beta signaling. This persistent activation contributes to the chronicity of the fibrotic process. The continued activation of TGF-beta/Smad signaling leads to an ongoing synthesis of extracellular matrix components, surpassing the normal tissue repair requirements. A positive feedback loop may develop, where the deposited extracellular matrix promotes further TGF-beta activation, perpetuating the fibrotic process.

In animal studies, UUO causes continuous pressure damage to the kidneys, accelerating fibrosis. In cellular studies, TGF-beta-induced changes in Smad2/3 occur relatively quickly and are known to be transient. Inman et al. reported that the levels of nuclear phospho-Smad2 reach their highest point around 45 minutes after TGF-β stimulation and thereafter decrease after 5 hours (REF 1). Cui et al. investigated that TGF-β provoked a rapid SMAD2 phosphorylation within 15 min. Phospho-Smad2 reached its maximum after 1 hr both under normoxia and hypoxia. After 4 h, the level of p-Smad2 gradually declined under hypoxia compared to normoxia. These differences became significant after 8 hr (REF 2). Sustained receptor activation is necessary to sustain the presence of active Smads in the nucleus and to facilitate TGF-β-induced transcription.

- REF 1. Inman GJ, Nicolás FJ, Hill CS. Nucleocytoplasmic Shuttling of Smads 2, 3, and 4 Permits Sensing of TGF-β Receptor Activity. Mol Cell. 2002;10(2):283-94. doi: 10.1016/s1097-2765(02)00585-3

- REF 2. Cui W, Zhou J, Dehne N, Brüne B. Hypoxia induces calpain activity and degrades SMAD2 to attenuate TGFβ signaling in macrophages. Cell Biosci. 2015:5:36. doi: 10.1186/s13578-015-0026-x. eCollection 2015

In summary, TGF-β/Smad signaling is involved in the early initiation of fibrosis and contributes to the progression of fibrosis through sustained activation. Cellular experiments have shown that activation of phospho-Smad occurs between 45 minutes and 1 hour after stimulation and disappears after 4-5 hours. In this study, stimulation of HK-2 cells with TGF-β resulted in a decrease in Smad2/3 activation at 48 hours, but we observed a significant increase in the 6-hour early model. Therefore, to observe Smad activation in cellular models, it is necessary to use an early model within 6 hours. The understanding of these differences is crucial for developing in vitro model and targeted therapeutic strategies to intervene in the fibrotic process at different stages.

The authors added these explanation as follows: “(line 321-332) The response of TGF-β/Smad signaling and phospho-Smad2/3 can differ between the early and late stages of fibrosis. In the early stages of fibrosis, the TGF-β/Smad signaling pathway promotes the activation of fibroblasts, leading to the production and deposition of extracellular matrix components. In late fibrosis, there is often sustained and prolonged activation of TGF-β signaling. This persistent activation contributes to the chronicity of the fibrotic process. In animal studies, UUO causes continuous pressure damage to the kidneys, accelerating fibrosis. Cellular experiments have shown that activation of phospho-Smad2 occurs between 45 minutes and 1 hour after stimulation and disappears after 4-5 hours [33, 34]. In this study, stimulation of HK-2 cells with TGF-β resulted in a decrease in Smad2/3 activation at 48 hours, but we observed a significant increase in the 6-hour early model. Therefore, to observe Smad activation in cellular models, it is necessary to use an early model within 6 hours. The understanding of these differences is crucial for developing in vitro model and targeted therapeutic strategies to intervene in the fibrotic process at different stages.”

1-2. Thank you for the valuable comment. Following the reviewer's comments, we have re-quantified the western blot gels for TGF-beta/Smad pathway-related markers (Fig. 3). We apologize for the inaccurate results and figures. The text description for Fig. 3 has also been modified as follows: “(line 190-198) TGF-β treatment for 6-h increased the expression of phospho-Smad2/3 (1.64 ± 0.06 fold changes compared to control, P < 0.001) and Smad ubiquitination regulator 1 (Smurf1) (1.46 ± 0.15 fold changes compared to control, P < 0.001), which was significantly reduced by MN705 treatment (Fig. 3A, B). In the TGF-β-treated (48-h) HK-2 cell in vitro model, Smurf1 expression increased and significantly attenuated by MN705 treatment (1.70 ± 0.35, 1.48 ± 0.24, 0.65 ± 0.24, and 0.18 ± 0.19 fold changes compared to control, overall P < 0.001; TGF-β 2 ng/ml with vehicle, MN705 0.05 mg/ml, 0.01 mg/ml, and 0.02 mg/ml, respectively); however, phospho-Smad2/3 decreased with prolonged exposure to TGF-β and the decrease after MN705 treatment was not significant (Fig. 3C, D).”

1-3. Thank you for your valuable suggestion. Rubus coreanus, a natural substance, consists of a variety of ingredients. Analyzing the constituents of Rubus coreanus and identifying those that have antifibrotic effects is important, but it requires additional experiments that are beyond the scope of this study. Therefore, the authors added the following information from the literature review to the discussion section: “(line 283-The constituents of Rubus coreanus can vary, but it generally contains a range of bioactive compounds that contribute to its potential health benefits. The representative components of Rubus coreanus include polyphenols, vitamins, and minerals. Polyphenolic compounds, such as anthocyanins, flavonoids, and ellagic acid have been widely studied for their antioxidant properties. Rubus coreanus may contain various vitamins and minerals, such as vitamin C and manganese, which also contribute to its antioxidant capacity. The antioxidant properties of Rubus coreanus can play a crucial role in neutralizing free radicals and reducing the production of reactive oxygen species and pro-inflammatory cytokines, thereby inhibiting fibrogenesis [21].”

“(line291-298) There has been research investigating the potential role of antioxidants in treating kidney fibrosis in experimental studies. Antioxidants are studied for their ability to reduce oxidative stress, inflammation, and cellular damage, which are associated with the development of fibrosis. N-Acetylcysteine (NAC) is a precursor to glutathione, a powerful endogenous antioxidant. Some studies have investigated the potential of NAC and it’s anti-oxidative effect in ameliorating kidney fibrosis [22]. Resveratrol is a polyphenol found in certain foods, including red grapes and berries. Chowdhury et al. demonstrated that resveratrol treatment reduced kidney fibrosis in a high-fat diet rat model, possibly through its antioxidant and anti-inflammatory effects [23].”

“(line 345-347) While these studies suggest a potential therapeutic role for antioxidants in kidney fibrosis, it's essential to approach the findings with caution.” The translation of experimental results to clinical applications requires further research, and “(line 347-348) the effectiveness of antioxidants may vary depending on the specific conditions and causes of kidney fibrosis.” It's important to note that while Rubus coreanus shows promise in terms of its antioxidant properties, “(line 350-351) further research is needed to fully understand the specific mechanisms and potential health benefits associated with Rubus coreanus.”

- [22] Shen Y, Miao NJ, Xu JL, Gan XX, Xu D, Zhou L, et al. N-acetylcysteine alleviates angiotensin II-mediated renal fibrosis in mouse obstructed kidneys. Acta Pharmacol Sin. 2016;37(5):637-44. doi: 10.1038/aps.2016.12

- [23]. Chowdhury FI, Yasmi T, Akter R, Islam MN, Hossain MM, Khan F, et al. Resveratrol treatment modulates several antioxidant and anti-inflammatory genes expression and ameliorated oxidative stress mediated fibrosis in the kidneys of high-fat diet-fed rats. Saudi Pharm J. 2022;30(10):1454-1463. doi: 10.1016/j.jsps.2022.07.006

1-4. Thank you for the insightful comment. In general, renal fibrosis is an intractable disease, making it difficult to prove the efficacy of treatments. Furthermore, natural products are typically used for health promotion and disease prevention rather than as therapeutic agents. In light of these considerations, the researchers concluded that the null result was likely to be confirmed if the MN705 used in this study was used for therapeutic purposes. In response to the reviewer's good comments, we have added the following additional statement to the discussion: "(line 349-351) In addition, this study only confirmed the preventive effect of Rubus coreanus on anti-fibrosis. Further experiments and clinical studies are necessary to determine if the Rubus coreanus has a therapeutic effect on kidney fibrosis.”

2-1/2/3. Thank you for the valuable comment. In the original submission, the researchers wanted to show that the MN705 compound had antifibrotic effects in animal models and to explain how these effects happened in the TGF-β-challenged HK-2 kidney cellular models with TGF- β/Smad pathway and oxidative stress. We could confirm that MN705 attenuated the kidney cellular fibrosis response and increased antioxidants in the 48-hour model (Fig. 2). However, we could not confirm the expected Smad pathway change in 48-hour model. This led the researchers to check the 6-hour early model, thinking that the Smad pathway that induces fibrosis might act earlier than 48 hours. The 6-hour model showed what was expected: it proved the changes in the Smad pathway (increased Smurf1 and phospho-Smad2/3 decrease after MN705) that could explain why renal cell fibrosis was prevented after MN705 treatment.

A substantial research supports the role of oxidative damage as a cause of kidney fibrosis. The purpose of in vitro experiment was not to prove that oxidative damage causes kidney fibrosis but to create a model of kidney cell fibrosis that can mimic the animal model of kidney fibrosis. Therefore, TGF-β was challenged to HK-2 cells.

In our original submission, we did not find any evidence of anti-oxidation or changes in markers of oxidative stress along with improvement of fibrosis in the animal in vivo model. To correct this discrepancy, we investigated the changes in MnSOD and 8-OHDG in the animal model (Fig. 7). Following MN705 treatment, MnSOD (anti-oxidative enzyme), which was reduced in UUO, showed a tendency to increase. The oxidative stress of 8-OHDG was significantly increased in UUO and significantly decreased after MN705 treatment. These results were described as follows: “(line 234-240) In addition, we also identified changes in anti-oxidative enzymes and oxidative stress in an in vivo model. The expression of MnSOD decreased significantly after the UUO operation. In the MN705 treatment group, reduced MnSOD expression showed a relatively increased response with no statistical significance. We measured the level of oxidative stress using 8-OHDG immunohistochemical staining. The UUO group showed significantly increased tissue expression levels of 8-OHDG. After MN705 treatment, the MN705 treatment group showed a significant decrease in oxidative stress expressed with 8-OHDG.”

2-4. Thank you for the valuable comment. Although not detailed in the paper, various studies were consulted to determine the animal dose of Rubus coreanus. In the studies described below, we found that doses of 50-500 mg/kg/day were commonly used as therapeutic doses. Lee et al. reported anti-obesity effects at a therapeutic dose of 300 mg/kg/day, and Kim et al. confirmed anti-dyslipidemic effects at 250 mg/kg/day. In general, antifibrotic effects are not easy to prove experimentally. Therefore, we decided to use a relatively high dose of 300 mg/kg/day as the primary dose to prevent the null results. In addition, we set up a 600 mg/kg/day high dose group in this experiment to check the dose response of antifibrotic effects and other unexpected side effects.

- Om et al. Molecules. 2016;21(1):65. doi: 10.3390/molecules21010065

- Oh et al. Evid Based Complement Alternat Med. 2016:2016:4357656. doi: 10.1155/2016/4357656

- Jeong et al. Int J Obes (Lond). 201;39(3):456-64. doi: 10.1038/ijo.2014.155

- Jung et al. J Med Food. 2007;10(4):689-93. doi: 10.1089/jmf.2006.006

- Lee et al. Kor J Food Nutrition. 2018;31(2):242-51. doi:10.9799/ksfan.2018.31.2.242

- Kim et al. Nutrients. 2018 1;10(12):1846. doi: 10.3390/nu10121846.

2-5. Thank you for the critical comment. As the reviewer notes, the response of MN705 may be different in animal studies, as shown in cellular experiments with early versus late stages of fibrosis. Recently, many researchers in the nephrology field have shown great interest in exploring the pathophysiological characteristics of a new disease entity called the AKI-to-CKD transition. This kidney disease model progresses to fibrosis after an acute kidney injury. Researchers have widely used the unilateral ischemia-reperfusion injury model as an animal model of the AKI-to-CKD transition. However, this study explored the anti-fibrotic effect of MN705 in a renal fibrosis model and its mechanism. In the UUO model, fibrosis is typically identified between days 7 and 14; in this study, we observed fibrosis specifically at day 7. It would be intere

---

## [Decision Letter · Decision Letter 1]

18 Apr 2024

PONE-D-23-21717R1Rubus coreanus extract attenuates kidney fibrosis through TGF-β/Smad pathway inhibitionPLOS ONE

Dear Dr. Lee,

Thank you for submitting your manuscript to PLOS ONE. After careful consideration, we feel that it has merit but does not fully meet PLOS ONE’s publication criteria as it currently stands. Therefore, we invite you to submit a revised version of the manuscript that addresses the points raised during the review process.

We look forward to receiving your revised manuscript.

Kind regards,

Qiang Ding, Ph.D.

Academic Editor

PLOS ONE

Journal Requirements:

Reviewers' comments:

Reviewer's Responses to Questions

**Comments to the Author**

1. If the authors have adequately addressed your comments raised in a previous round of review and you feel that this manuscript is now acceptable for publication, you may indicate that here to bypass the “Comments to the Author” section, enter your conflict of interest statement in the “Confidential to Editor” section, and submit your "Accept" recommendation.

Reviewer #1: All comments have been addressed

Reviewer #2: (No Response)

2. Is the manuscript technically sound, and do the data support the conclusions?

Reviewer #1: Yes

Reviewer #2: No

3. Has the statistical analysis been performed appropriately and rigorously? 

Reviewer #1: Yes

Reviewer #2: N/A

4. Have the authors made all data underlying the findings in their manuscript fully available?

Reviewer #1: Yes

Reviewer #2: Yes

5. Is the manuscript presented in an intelligible fashion and written in standard English?

Reviewer #1: Yes

Reviewer #2: Yes

6. Review Comments to the Author

Reviewer #1: The revised manuscript appropriately addressed all the raised concerns regardiing in vitro and in vivo experimentation.

No more comments

Reviewer #2: There are some major weaknesses to be addressed. One is the model is preventative, don't support its role in treatment of kidney fibrosis, it should be evaluated after kidney fibrosis established. Otherwise, its best role is preventative, not treatment.

Even preventative, the data has major flaws. HK2 cells, not primary cells used for the tested. The drug kinetics and time- dose-effects are not evaluated, significantly reduced the confidence of the research.

7. PLOS authors have the option to publish the peer review history of their article (what does this mean? ). If published, this will include your full peer review and any attached files.

**Do you want your identity to be public for this peer review?** For information about this choice, including consent withdrawal, please see our Privacy Policy .

Reviewer #1: No

Reviewer #2: No

---

## [Author Response · Author response to Decision Letter 2]

14 Jul 2024

Response to the Editor: The authors appreciate the decision and comments from the editor and reviewers. important comment. The authors have made the necessary improvements. In this revision, we have further addressed the implication and limitation of this study. Thank you again for the time and opportunity for this revision. Through the revision process, the authors were able to fill in the gaps and provide a more logical explanation for our findings.

The required files of a rebuttal letter (file labeled 'Response to Reviewers'), a marked-up copy of the manuscript (file labeled 'Revised Manuscript with Track Changes'), and an unmarked version of the revised paper without tracked changes ('Manuscript') were all prepared and uploaded. Financial disclosures are newly added to the editorial manager system. The authors have endeavored to follow the guidelines for resubmitting figure files. The authors ensure that this manuscript meets PLOS ONE's style requirements.

With this revision, we hope to have this study published in PLOS ONE. Thank you very much.

Response to the Reviewer 2: Thank you for the important comment. I agree with the reviewer's point. In this study, Rubus coreanus extract was administered before the development of fibrosis, and therefore, the preventive effect (not therapeutic effect) of Rubus coreanus extract was verified in this study. Therefore, we have revised the title of the manuscript from "Rubus coreanus extract attenuates kidney fibrosis through TGF-β/Smad pathway inhibition" to "Rubus coreanus extract prevents kidney fibrosis through TGF-β/Smad pathway inhibition".

In the abstract, “we aimed to investigate the efficacy of water-soluble extract of Rubus coreanus (MN705) in attenuating kidney fibrosis in a mouse model of unilateral ureteral obstruction (UUO) and in an in vitro model of TGF-β-challenged HK-2 cells.” > “we aimed to investigate the efficacy of water-soluble extract of Rubus coreanus (MN705) in preventing kidney fibrosis in a mouse model of unilateral ureteral obstruction (UUO) and in an in vitro model of TGF-β-challenged HK-2 cells.”; “a potential target for treatment of kidney fibrosis.” > “a potential target for prevention of kidney fibrosis.

In the discussion, we also stated that “In addition, this study only confirmed the preventive effect of Rubus coreanus on anti-fibrosis. Further experiments and clinical studies are necessary to determine if the Rubus coreanus has a therapeutic effect on kidney fibrosis.”

At this stage of revision, it is beyond the capabilities of my lab to replicate this experimental study with primary kidney tubular cells rather than HK2 cells. Therefore, I have included these limitations in the Discussion section as follows, “This study showed the effects of Rubus coreanus only in the UUO mice kidney fibrosis model and TGF-β challenged HK2 cellular model, which may be a limitation in generalizing the results of this study.”

In this study, the dose-dependent effect of MN705 (Rubus coreanus extract) was demonstrated in cellular and animal models, as shown in Fig. 2, Fig. 3 (in vitro, MN705 0.05, 0.1, 0.2 mg/ml), and Fig. 4, Fig. 5, Fig. 6 (in vivo, MN705 300, 600 mg/day). In cellular experiments, a dose-dependent improvement of fibrosis markers (fibronectin, collagen1) and an increase in antioxidants (MnSOD) were observed (Fig. 2). Animal studies also confirmed that alpha-SMA, collagen1 fibrosis markers decreased at high concentrations and pSmad 2/3 decreased in a concentration-dependent manner with increasing doses of MN705. 8-OHDG, a marker of oxidative damage, also decreased significantly with increasing MM705 concentration.

Although the study has some limitations, this is a meaningful study that demonstrates the protective effect of MN705 (Rubus coreanus extract) against kidney fibrosis in cellular and animal models. Especially, this study has advantages since the results were improved by conducting revisions as pointed out by the editors and reviewers. We hope that this revision will give this study the opportunity to be published in the Plos One.

---

## [Decision Letter · Decision Letter 2]

9 Aug 2024

PONE-D-23-21717R2Rubus coreanus extract prevents kidney fibrosis through TGF-β/Smad pathway inhibitionPLOS ONE

Dear Dr. Lee,

Thank you for submitting your manuscript to PLOS ONE. After careful consideration, we feel that it has merit but does not fully meet PLOS ONE’s publication criteria as it currently stands. Therefore, we invite you to submit a revised version of the manuscript that addresses the points raised during the review process.

**Reviewer is not satisfied with the response. Detailed comments included. Please address the reviewer's comments accordingly. **

We look forward to receiving your revised manuscript.

Kind regards,

Qiang Ding, Ph.D.

Academic Editor

PLOS ONE

Journal Requirements:

Reviewers' comments:

Reviewer's Responses to Questions

**Comments to the Author**

1. If the authors have adequately addressed your comments raised in a previous round of review and you feel that this manuscript is now acceptable for publication, you may indicate that here to bypass the “Comments to the Author” section, enter your conflict of interest statement in the “Confidential to Editor” section, and submit your "Accept" recommendation.

Reviewer #2: (No Response)

2. Is the manuscript technically sound, and do the data support the conclusions?

Reviewer #2: Partly

3. Has the statistical analysis been performed appropriately and rigorously? 

Reviewer #2: No

4. Have the authors made all data underlying the findings in their manuscript fully available?

Reviewer #2: Yes

5. Is the manuscript presented in an intelligible fashion and written in standard English?

Reviewer #2: Yes

6. Review Comments to the Author

Reviewer #2: Congratulations for author’s arguments that there are difficulties in addressing the concerns. Hoowever, here are minimal requirements in order to meet with review criteria.

1. The limitation added to the discussion is very short and too brief, and should be enough for readers to understand the limitation of the study. It needs to include specifics such as that this work is not done in primary cells and there are limitations for this.

2. This reviewer understands that authors may not be able to replicate all or most of the data in primary cells at this stage, but, it is necessary to replicate or confirm some key findings, even one or two results, to satisfy the minimal requirement. This is minimal requirement as this stage. Otherwise, the scientific value is totally not convinced.

7. PLOS authors have the option to publish the peer review history of their article (what does this mean? ). If published, this will include your full peer review and any attached files.

**Do you want your identity to be public for this peer review?** For information about this choice, including consent withdrawal, please see our Privacy Policy .

Reviewer #2: No

---

## [Author Response · Author response to Decision Letter 3]

31 Oct 2024

Response to the Editor: The authors appreciate the decision and comments from the editor and reviewers’ important comments. The authors have made the necessary improvements. In this revision, we performed in vitro tests with human primary renal proximal tubule epithelial cells and validated the findings from the HK-2 cell line presented in the original manuscript. In addition, we have further addressed the implication and limitation of this study. Thank you again for the time and opportunity for this revision. Through the revision process, the authors were able to fill in the gaps and provide a more logical explanation for our findings.

The required files of a rebuttal letter (file labeled 'Response to Reviewers'), a marked-up copy of the manuscript (file labeled 'Revised Manuscript with Track Changes'), and an unmarked version of the revised paper without tracked changes ('Manuscript') were all prepared and uploaded. Financial disclosures are included in the editorial manager system. The authors have endeavored to follow the guidelines for resubmitting figure files. The authors ensure that this manuscript meets PLOS ONE's style requirements.

With this revision, we hope to have this study published in PLOS ONE. Thank you very much.

Response to the Reviewer #2-1: Thank you for the important comment. I agree with the reviewer's point. In response to the reviewer's constructive suggestions for enhancing this research, we performed additional in vitro tests using human renal proximal tubular epithelial cells (RPTEC, Lonza, Catalog #: CC-2553).

In an in vitro test using RPTEC, we were able to confirm the results obtained with HK-2 cells. The antioxidant MnSOD increased after MN705 treatment, while fibrosis-associated Smad2/3, phospho-smad2/3, and fibronectin decreased. The results are incorporated as Figure 4 and detailed as follows: “The results found in the HK-2 cell line were validated using human renal proximal tubular epithelial cells. After MN705 treatment, the epithelial cell morphology altered by TGF-β 2 ng/ml for 48-h was restored to a more elongated and spindle-shaped morphology (Fig. 4A). Western blots were employed to investigate alterations of proteins associated with anti-oxidation and kidney fibrosis. MN705 led to enhanced expression of the antioxidant MnSOD, reduced fibronectin levels, and diminished phospho-Smad2/3 (Fig. 4B, C).”

In addition, we describe the implications and limitations of our study in more detail as follows: “This study has the unique advantage of being the first to show the antifibrotic properties of Rubus coreanus in kidney fibrosis. Additionally, it proposes that the modulation of TGF-β/Smad signaling, along with the anti-oxidative effects, is the underlying mechanism of anti-fibrosis. In addition, the effects of Rubus coreanus on TGF-β/Smad signaling, antioxidants, and preventing fibrosis were confirmed in both animal study and two in vitro models. While these studies suggest a potential therapeutic role for antioxidants in kidney fibrosis, it's essential to approach the findings with caution. The effectiveness of antioxidants may vary depending on the specific conditions and causes of kidney fibrosis. The UUO model induces kidney fibrosis through a sustained pressure effect within the kidney. This pathophysiological trait contrasts with most clinical disorders associated with kidney fibrosis. In addition, TGF-β is a principal pro-fibrotic factor that activates myofibroblasts and serves as a crucial mediator in the progression of kidney fibrosis. However, it is essential to experimentally replicate many circumstances associated with kidney fibrosis, including hypoxia, infection, and inflammation, to facilitate the generalization of experimental results. In addition, this study only confirmed the preventive effect of Rubus coreanus on anti-fibrosis. Further experiments and clinical studies are necessary to determine if the Rubus coreanus has a therapeutic effect on kidney fibrosis. At last, the anti-fibrotic impact of Rubus coreanus was only observed in the 7-day model of the UUO mouse. Therefore, it is worthwhile to investigate if this anti-fibrotic effect persists over shorter or longer time intervals.”

Response to the Reviewer #2-2: Thank you for the critical and valuable comment. I totally agree with the reviewer's point. Previously, we utilized HK-2 cells, a representative renal tubule cell line, to demonstrate the mechanism of the anti-fibrotic effect of Rubus coreanus in the kidney, focusing on the time-dependent alterations in the Smad pathway during TGF-β-induced fibrosis. Nonetheless, owing to the disparities between animal and human kidney cells, one may doubt the applicability of results obtained from animal cells to human. In response to the reviewer's constructive suggestions for enhancing this research, we procured human renal proximal tubular epithelial cells (RPTEC, Lonza, Catalog #: CC-2553) and performed additional in vitro studies.

In an in vitro test using RPTEC, we were able to confirm the results obtained with HK-2 cells. The antioxidant MnSOD increased after MN705 treatment, while fibrosis-associated Smad2/3, phospho-smad2/3, and fibronectin decreased. The results are incorporated as Figure 4 and detailed as follows: “The results found in the HK-2 cell line were validated using human renal proximal tubular epithelial cells. After MN705 treatment, the epithelial cell morphology altered by TGF-β 2 ng/ml for 48-h was restored to a more elongated and spindle-shaped morphology (Fig. 4A). Western blots were employed to investigate alterations of proteins associated with anti-oxidation and kidney fibrosis. MN705 led to enhanced expression of the antioxidant MnSOD, reduced fibronectin levels, and diminished phospho-Smad2/3 (Fig. 4B, C).”

We thank the editors and reviewers for their important comments during the three revisions, which improved the paper. We hope that this revision will give this study the opportunity to be published in the Plos One.

---

## [Decision Letter · Decision Letter 3]

27 Dec 2024

PONE-D-23-21717R3Rubus coreanus extract prevents kidney fibrosis through TGF-β/Smad pathway inhibitionPLOS ONE

Dear Dr. Lee,

Thank you for submitting your manuscript to PLOS ONE. After careful consideration, we feel that it has merit but does not fully meet PLOS ONE’s publication criteria as it currently stands. Therefore, we invite you to submit a revised version of the manuscript that addresses the points raised during the review process.

Reviewer raised serious question about the conflict interest for your revision, in addition to the scientific data supporting the conclusion, and data interpretation and if sufficient consideration has been given to the potential confounding factors that may affect the supporting data and conclusion.

We look forward to receiving your revised manuscript.

Kind regards,

Qiang Ding, Ph.D.

Academic Editor

PLOS ONE

Reviewers' comments:

Reviewer's Responses to Questions

**Comments to the Author**

1. If the authors have adequately addressed your comments raised in a previous round of review and you feel that this manuscript is now acceptable for publication, you may indicate that here to bypass the “Comments to the Author” section, enter your conflict of interest statement in the “Confidential to Editor” section, and submit your "Accept" recommendation.

Reviewer #2: (No Response)

2. Is the manuscript technically sound, and do the data support the conclusions?

Reviewer #2: Partly

3. Has the statistical analysis been performed appropriately and rigorously? 

Reviewer #2: No

4. Have the authors made all data underlying the findings in their manuscript fully available?

Reviewer #2: Yes

5. Is the manuscript presented in an intelligible fashion and written in standard English?

Reviewer #2: Yes

6. Review Comments to the Author

Reviewer #2: The authors have declared that no competing interests exist. However, this revision has serous conflict interest to be addressed.

This revision has included the company, Medvill Co., Ltd., which provides the testing drug/ Rubus coreanus extract extract powders used for the study, and this company was not included in any of the prior versions. This is a serious issue. One is that providing material is not warranty an authorship, based on the journal guideline. Authors’ contribution is vague and no specifics for why the company is added to the paper. Second, most importantly, conflict interest. The data are now considered directly related to the company benefit and the results and adding the company as the authorship promote the company product and image by this manuscript. Therefore, these are potential conflict interest, and the results have to be re-evaluated as a promoting manuscript for the company’s product, rather than only a purely scientific research paper with insufficient data to support the conclusion.

Giving the nature of this manuscript is therapeutic intervention, this could serve as an advertisement for the company, a serious conflict interest, as the company is added at the last minute, and this version and last version only have minimal difference between them.

As these changes, this reviewer would like to required more in vivo evidence for the prevention. Current results only indicates reduction of some pathological changes.

Mostly importantly, the figures and labels are low quality, unclear, difficult to judge.

Limited animal data to support the conclusion. Figs 5-7 are from the animal studies, figures are low quality and unclear. The connection between the figures are not studied.

Major weaknesses as following:

1. There is no evaluation of pharmacologic toxicity in animals

2. the internal drug amount such as concentration in blood and kidney tissues.

3. What are the mechanism for the MN705 reduces TGFbeta signaling and matrix deposition?

4. Is the drug MN705 binds indirectly or directly to TGFbeta receptor, or what are the action route?

5. Why NN705, is it better than using anti-TGFbeta inhibitors directly?

6. Inhibition of TGFbeta may raise cancer concern. How about MN705, any concern, if only targeting TGFbeta?

7. is MN705 permeable to cells and kidney tissues?

8. How the MN705 affects MnSOD?

9. And how MnSOD is related to TGFbeta signaling, and UUO kidney fibrosis?

10. Is MnSOD increase by MN705 a result or critical pathway?

11. What if MnSOD is blocked, and animal are treated with MN705, will kidney fibrosis changes during these conditions?

12. Why Fig.6A missing one lane of proteins for “UUO+MN705 600mg” condition? Is the experiment only done once?

13. Also, Fig.6 does not show much diffierence for pSmad2/3 levels between UUO and UUO+MN705, but the fibrosis is difference. Any explanation?

7. PLOS authors have the option to publish the peer review history of their article (what does this mean? ). If published, this will include your full peer review and any attached files.

**Do you want your identity to be public for this peer review?** For information about this choice, including consent withdrawal, please see our Privacy Policy .

Reviewer #2: No

---

## [Author Response · Author response to Decision Letter 4]

25 Jan 2025

Response: The authors appreciate the decision and comments from the editor and reviewers’ important comments. During the three rounds of revisions, we diligently addressed the editor's and reviewers' requests by supplementing the experimental content and preparing thorough responses. During the third revision process, an investigator from the company that provided the natural product conducted the proofreading of the manuscript. This individual had initially provided the natural product and participated in the early stages of the research purely for academic purposes. Furthermore, she also contributed to the cell experiments added during the revision process. Based on these contributions and with the consent of the other authors, we deemed the investigator qualified for authorship and added she as a co-author.

However, the reviewer pointed out that adding an author with a potential conflict of interest during the revision process could raise significant ethical concerns. All authors agree that including an investigator from the material-supplying company as a co-author and explicitly indicating their affiliation with the company may conflict with ethical standards regarding conflicts of interest. Therefore, to demonstrate that the natural product was provided solely for academic purposes and to ensure there are no conflicts of interest, we have decided to exclude the company and its representative from the list of authors with consents of all authors. We kindly ask for your understanding regarding the additional critique raised by the reviewer, as it pertains to new research directions that are difficult to pursue at this stage of the study.

Thank you again for the time and opportunity for this revision. Through the revision process, the authors were able to fill in the gaps and provide a more logical explanation for our findings. The required files of a rebuttal letter (file labeled 'Response to Reviewers'), a marked-up copy of the manuscript (file labeled 'Revised Manuscript with Track Changes'), and an unmarked version of the revised paper without tracked changes ('Manuscript') were all prepared and uploaded. Financial disclosures are included in the editorial manager system. The authors have endeavored to follow the guidelines for resubmitting figure files. The authors ensure that this manuscript meets PLOS ONE's style requirements.

With this revision, we hope to have this study published in PLOS ONE. Thank you very much.

Response: Thank you for the important comment. I agree with the reviewer's point. During the three rounds of revisions, we diligently addressed the editor's and reviewers' requests by supplementing the experimental content and preparing thorough responses. During the third revision process, an investigator from the company that provided the natural product conducted the proofreading of the manuscript. This individual had initially provided the natural product and participated in the early stages of the research purely for academic purposes. Furthermore, they contributed to the cell experiments added during the revision process. Based on these contributions and with the consent of the other authors, we deemed the investigator qualified for authorship and added she as a co-author.

However, the reviewer pointed out that adding an author with a potential conflict of interest during the revision process could raise significant ethical concerns. All authors agree that including an investigator from the material-supplying company as an author and explicitly indicating their affiliation with the company may conflict with ethical standards regarding conflicts of interest. Therefore, to demonstrate that the natural product was provided solely for academic purposes and to ensure there are no conflicts of interest, we have decided to exclude the company and its representative from the list of authors. We kindly ask for your understanding regarding the additional critique raised by the reviewer, as it pertains to new research directions that are difficult to pursue at this stage of the study.

In addition, the reviewer also pointed out that the quality of the presented images is not satisfactory. Upon reviewing the images included in the PDF, we agree that the resolution appears low. However, when viewing the original images we provided for download, the high-resolution versions are clearly visible. If necessary, we can submit individual files for each image separately.

Response to Q1. This study primarily aims to investigate the effects of Rubus coreanus extract on kidney disease. In the cellular model, toxicity was assessed using the MTT assay. While there is a potential risk of toxicity at higher concentrations in in vivo experiments, no toxicity was observed within the design of this study. Although the possibility of toxicity at high concentrations exists, this study holds value as a report focused on verifying the efficacy of Rubus coreanus extract.

Response to Q2. Rubus coreanus extract is a natural compound complex composed of various components, making it challenging to measure its concentration in blood and kidney tissues.

Response to Q3. Thank you for important comments. As presented in the results of this study, MN705 appears to alleviate fibrosis by regulating the TGF-β/Smad pathway and mitigating oxidative damage. Further studies are needed to elucidate the underlying mechanisms by which MN705 regulates the TGF-β/Smad pathway and reduces oxidative damage.

Response to Q4. Thank you for important comments. Rubus coreanus extract is a natural compound complex composed of various components. Therefore, further studies are needed to identify which specific components bind to the TGF-β receptor and to elucidate the mechanisms by which they mediate signaling.

Response to Q5. In this study, we validated the anti-fibrotic effects of Rubus coreanus extract in a kidney disease model but did not compare its effects with other TGF-β inhibitors. Comparative studies with other TGF-β inhibitors should be conducted in future research.

Response to Q6. As previously explained, Rubus coreanus extract is a natural compound complex composed of various components. It is thought to alleviate fibrosis in kidney disease models through multiple mechanisms. In this study, we focused on elucidating its mechanisms, particularly in regulating TGF-β signaling and mitigating oxidative damage. Many therapeutic agents currently used for chronic diseases provide significant benefits in alleviating disease but may also have associated side effects. As a natural compound complex, Rubus coreanus extract is noteworthy for its potential to offer beneficial biological effects with relatively fewer side effects. Beyond the TGF-β signaling and oxidative damage mitigation mechanisms identified in this study, other potential mechanisms warrant further investigation in future research.

Response to Q7. As previously explained, Rubus coreanus extract is a natural compound complex composed of various components. It is thought to alleviate fibrosis in kidney disease models through multiple mechanisms. In this study, we focused on elucidating its mechanisms, particularly in regulating TGF-β signaling and mitigating oxidative damage. Many therapeutic agents currently used for chronic diseases provide significant benefits in alleviating disease but may also have associated side effects. As a natural compound complex, Rubus coreanus extract is noteworthy for its potential to offer beneficial biological effects with relatively fewer side effects. Beyond the TGF-β signaling and oxidative damage mitigation mechanisms identified in this study, other potential mechanisms warrant further investigation in future research.

Response to Q8. Thank you for important comments. As presented in the results of this study, MN705 appears to alleviate fibrosis by regulating the TGF-β/Smad pathway and mitigating oxidative damage. Further studies are needed to elucidate the underlying mechanisms by which MN705 reduces oxidative damage via MnSOD.

Response to Q9. This study aimed to investigate the anti-fibrotic effects of MN705 in a kidney disease model, and as such, the mechanisms by which MnSOD influences TGF-β signaling and fibrosis fall outside the scope of this research. However, relevant insights can be drawn from prior studies. Increased oxidative stress can activate TGF-β signaling, leading to fibrosis and tissue remodeling in various diseases. Importantly, high MnSOD activity can counteract oxidative stress, thereby inhibiting excessive TGF-β activation and potentially preventing fibrosis development (J Gerontol A Biol Sci Med Sci. 2015;70(5):533. doi: 10.1093/gerona/glu090). Additionally, studies have demonstrated that increased MnSOD expression or activity can suppress TGF-β signaling by reducing oxidative stress, potentially limiting fibrosis development in various tissues (Int J Mol Sci. 2022;23(24):15893. doi: 10.3390/ijms232415893).

Response to Q10 and Q11. Based on the results of this study, it is inferred that MN705 is directly involved in the upregulation of MnSOD. The anti-fibrotic effects of MN705 may be attenuated when MnSOD is inhibited. However, as MN705 is a natural extract composed of various components, there is also a possibility that its anti-fibrotic effects are mediated through other mechanisms. Further studies are required to determine which specific components of Rubus coreanus extract contribute to its anti-fibrotic effects in kidney disease and through which mechanisms these effects are achieved.

Response to Q12: Thanks for your comment. As described in the Methods, male C57BL/6 mice (7 weeks old) were randomly assigned to the following groups: sham/vehicle (distilled water, n = 4), sham/MN705 (600 mg/kg/day, n = 4), UUO/vehicle (n = 6), UUO/MN705-low dose (300 mg/kg/day, n = 6), and UUO/MN705-high dose (600 mg/kg/day, n = 6). Western blot analysis was conducted in two separate sets, with the samples divided as follows: sham/vehicle (2 lanes), sham/MN705 (2 lanes), UUO/vehicle (3 lanes), UUO/MN705-low dose (3 lanes), and UUO/MN705-high dose (3 lanes). Our Figure 6A UUO+MN705 600mg did not lack lanes (3 lanes). And we had already attached the western bands that were only used for statistical purposes in the Supplements (western band raw, PAGE 15). All results presented in this study were validated through 2–3 repeated experiments.

Response to Q13: Thank you for important comment. The reviewer pointed out that the changes in pSmad presented in Fig. 6A do not appear significant. As previously explained, the western blot analysis was conducted in two separate sets, and statistical significance was confirmed when all the results were combined and analyzed. Please also review the additional western blot results provided in the supplementary materials.

We thank the editors and reviewers for their important comments during the three revisions, which improved the paper. We hope that this revision will give this study the opportunity to be published in the Plos One.

---

## [Decision Letter · Decision Letter 4]

4 Mar 2025

Rubus coreanus extract prevents kidney fibrosis through TGF-β/Smad pathway inhibition

PONE-D-23-21717R4

Dear Dr. Lee,

We’re pleased to inform you that your manuscript has been judged scientifically suitable for publication and will be formally accepted for publication once it meets all outstanding technical requirements.

Kind regards,

Qiang Ding, Ph.D.

Academic Editor

PLOS ONE

Additional Editor Comments (optional):

Reviewers' comments:

Reviewer's Responses to Questions

**Comments to the Author**

1. If the authors have adequately addressed your comments raised in a previous round of review and you feel that this manuscript is now acceptable for publication, you may indicate that here to bypass the “Comments to the Author” section, enter your conflict of interest statement in the “Confidential to Editor” section, and submit your "Accept" recommendation.

Reviewer #2: All comments have been addressed

2. Is the manuscript technically sound, and do the data support the conclusions?

Reviewer #2: Partly

3. Has the statistical analysis been performed appropriately and rigorously? 

Reviewer #2: N/A

4. Have the authors made all data underlying the findings in their manuscript fully available?

Reviewer #2: Yes

5. Is the manuscript presented in an intelligible fashion and written in standard English?

Reviewer #2: Yes

6. Review Comments to the Author

Reviewer #2: Thanks for addressing the comments and it is ok to leave some of the concerns for future research at this stage.

Revision addressed the potential COI, and to me, it is completely ok to include the person from the company and acknowledge the contribution of the company providing the materials for the studies, in the Acknowledge section of the manuscript.

7. PLOS authors have the option to publish the peer review history of their article (what does this mean? ). If published, this will include your full peer review and any attached files.

**Do you want your identity to be public for this peer review?** For information about this choice, including consent withdrawal, please see our Privacy Policy .

Reviewer #2: No

---

## [Editor Report · Acceptance letter]

PONE-D-23-21717R4

PLOS ONE

Dear Dr. Lee,

I'm pleased to inform you that your manuscript has been deemed suitable for publication in PLOS ONE. Congratulations! Your manuscript is now being handed over to our production team.

Kind regards,

on behalf of

Dr. Qiang Ding

Academic Editor

PLOS ONE